# Global genome diversity of the *Leishmania donovani* complex

Susanne U Franssen[1]*, Caroline Durrant[1], Olivia Stark[2], Bettina Moser[2], Tim Downing[1,3], Hideo Imamura[4], Jean-Claude Dujardin[4,5], Mandy J Sanders[1], Isabel Mauricio[6], Michael A Miles[7], Lionel F Schnur[8], Charles L Jaffe[8], Abdelmajeed Nasereddin[8], Henk Schallig[9], Matthew Yeo[7], Tapan Bhattacharyya[7], Mohammad Z Alam[10], Matthew Berriman[1], Thierry Wirth[11,12]*, Gabriele Schönian[2]*, James A Cotton[1]*

[1]Wellcome Sanger Institute, Wellcome Genome Campus, Hinxton, United Kingdom; [2]Charité Universitätsmedizin, Berlin, Germany; [3]Dublin City University, Dublin, Ireland; [4]Institute of Tropical Medicine, Antwerp, Belgium; [5]Department of Biomedical Sciences, University of Antwerp, Antwerp, Belgium; [6]Universidade Nova de Lisboa Instituto de Higiene e Medicina, Lisboa, Portugal; [7]London School of Hygiene and Tropical Medicine, London, United Kingdom; [8]Kuvin Centre for the Study of Infectious and Tropical Diseases, IMRIC, Hebrew University-Hadassah, Medical School, Jerusalem, Israel; [9]Amsterdam University Medical Centres – Academic Medical Centre at the University of Amsterdam, Department of Medical Microbiology – Experimental Parasitology, Amsterdam, Netherlands; [10]Department of Parasitology, Bangladesh Agricultural University, Mymensingh, Bangladesh; [11]Institut de Systématique, Evolution, Biodiversité, ISYEB, Muséum national d'Histoire naturelle, CNRS, Sorbonne Université, EPHE, Université des Antilles, Paris, France; [12]École Pratique des Hautes Études (EPHE), Paris Sciences & Lettres (PSL), Paris, France

**\*For correspondence:**
susanne.franssen@sanger.ac.uk (SUF);
thierry.wirth@mnhn.fr (TW);
gabriele.schoenian@t-online.de (GSö);
jc17@sanger.ac.uk (JAC)

**Competing interests:** The authors declare that no competing interests exist.

**Abstract** Protozoan parasites of the *Leishmania donovani* complex – *L. donovani* and *L. infantum* – cause the fatal disease visceral leishmaniasis. We present the first comprehensive genome-wide global study, with 151 cultured field isolates representing most of the geographical distribution. *L. donovani* isolates separated into five groups that largely coincide with geographical origin but vary greatly in diversity. In contrast, the majority of *L. infantum* samples fell into one globally-distributed group with little diversity. This picture is complicated by several hybrid lineages. Identified genetic groups vary in heterozygosity and levels of linkage, suggesting different recombination histories. We characterise chromosome-specific patterns of aneuploidy and identified extensive structural variation, including known and suspected drug resistance loci. This study reveals greater genetic diversity than suggested by geographically-focused studies, provides a resource of genomic variation for future work and sets the scene for a new understanding of the evolution and genetics of the *Leishmania donovani* complex.

## Introduction

The genus *Leishmania* is a group of more than 20 species of protozoan parasites that cause the neglected tropical disease leishmaniasis in humans, but also infect other mammalian hosts. Leishmaniasis is transmitted by phlebotomine sandflies and exists in four main clinical conditions: cutaneous leishmaniasis (CL), seen as single and multiple cutaneous lesions; mucocutaneous leishmaniasis (MCL), presenting in mucosal tissue; diffuse cutaneous leishmaniasis (DCL), seen as multiple nodular

cutaneous lesions covering much of the body; and visceral leishmaniasis (VL, also known as kala-azar), affecting internal organs. Disease incidence per year is estimated at 0.9 to 1.6 million new cases, mostly of CL, and up to 90,000 new cases per year of VL are associated with a 10% mortality rate (*Alvar et al., 2012*; *Burza et al., 2018*). The form of the disease is largely driven by the species of *Leishmania* causing the infection but is further influenced by vector biology and host factors, importantly by host immune status (*Burza et al., 2018*; *McCall et al., 2013*). In the mammalian host, parasites are intracellular, residing mainly in long lived macrophages. In the most severe visceral form, parasites infect the spleen, liver, bone marrow and lymph nodes, leading to splenomegaly and hepatomegaly. This results in a range of symptoms including frequent anaemia, thrombocytopenia and neutropenia, and common secondary infections which are often fatal without successful treatment (for review see: *Rodrigues et al., 2016*; *Burza et al., 2018*), although most infections remain asymptomatic (*Ostyn et al., 2011*).

The key species responsible for VL are *L. donovani* and *L. infantum* (see reviews *McCall et al., 2013*; *Burza et al., 2018*), which together form the *L. donovani* species complex. Both species mainly cause VL, but for each species atypical cutaneous presentations are common in some foci (reviewed in *Thakur et al., 2018*; for example, *Guerbouj et al., 2001*; *Zhang et al., 2014*). Post-kala-azar dermal leishmaniasis (PKDL), is a common sequel to VL that manifests with dermatological symptoms appearing after apparent cure of the visceral infection. PKDL is mainly seen on the Indian subcontinent and north-eastern and eastern Africa following infections caused by *L. donovani* (*Zijlstra et al., 2003*). *L. donovani* is considered to be largely anthroponotic even though the parasites can be encountered in animals (*Bhattarai et al., 2010*). In contrast, *L. infantum* – like most *Leishmania* species – causes a zoonotic disease, where dogs are the major domestic reservoir but a range of wild mammals can also be involved in transmission (*Díaz-Sáez et al., 2014*; *Quinnell and Courtenay, 2009*). Both species are widespread across the globe, with major foci in the Indian subcontinent and East Africa for *L. donovani*, the Mediterranean region and the Middle East for *L. infantum*, and China for both species (*Lun et al., 2015*; *Lysenko, 1971*; *Ready, 2014*). *L. infantum* has also more recently spread to the New World, via European migration during the 15th or 16th Century (*Leblois et al., 2011*), where it was sometimes described as a third species, *L. chagasi*. Leishmaniasis caused by parasites of the *L. donovani* complex differs across and even within geographical locations in the nature and severity of clinical symptoms (e.g. *Guerbouj et al., 2001*; *Zhang et al., 2014*; *Thakur et al., 2018*) and in the species of phlebotomine sandflies that act as vectors (*Alemayehu and Alemayehu, 2017*).

For this important human pathogen, there is a long history of interest in many aspects of the basic biology of *Leishmania*, including extensive interest in epidemiology, cell biology and immunology as well as the genetics and evolution of these parasites (e.g. *Simpson et al., 2006*; *Quinnell and Courtenay, 2009*; *Mougneau et al., 2011*). *Leishmania* has two unusual genomic features that influence its genetics, including mosaic aneuploidy and a complex and predominantly clonal life cycle. Aneuploidy is the phenomenon where individual chromosomes within a cell are of different copy numbers, and mosaic aneuploidy is where the pattern of chromosome dosage varies between cells of a clonal population (*Bastien et al., 1990*; *Sterkers et al., 2011*). Genome sequencing studies have shown extensive aneuploidy in cultured *Leishmania* field isolates (e.g. *Downing et al., 2011*; *Rogers et al., 2014*; *Zhang et al., 2014*; *Imamura et al., 2016*). Variation in chromosome dosage appears to be greater in in vitro than in vivo in animal models (*Dumetz et al., 2017*) or human tissues (*Domagalska et al., 2019*). However, these studies estimate average dosage of chromosomes in a population of sequenced cells. Only a few studies have directly investigated mosaicism between cells and these found it to be extensive both in vitro (*Sterkers et al., 2011*; *Lachaud et al., 2014*) and in vivo (*Prieto Barja et al., 2017*). Reproduction was originally thought to be predominantly clonal and this is still assumed to be the only mode of reproduction for the intracellular amastigotes found in the mammalian host. A number of studies have shown that hybridisation can occur during passage in the sandfly vector. This was demonstrated experimentally (e.g. *Akopyants et al., 2009*; *Romano et al., 2014*; *Inbar et al., 2019*) also showing evidence of meiosis (*Inbar et al., 2019*) and in field isolates through recombination-like signatures (*Cotton et al., 2019*; *Rogers et al., 2014*). However, the incidence of sexual reproduction in natural populations is still unclear (*Ramírez and Llewellyn, 2014*).

Despite this research, much remains unclear about the diversity, evolution and genetics of the *L. donovani* species complex. Difficult and laborious isoenzyme typing (*Rioux et al., 1990*) dominated

the description of *Leishmania* populations for at least 25 years (*Schönian et al., 2011*) but suffered from a critical lack of resolution, leading to convergent signals (*Jamjoom et al., 1999*). More recent typing schemes, using variation at small numbers of genetic loci (multi-locus sequence typing, MLST) or microsatellite repeats (multi locus microsatellite typing, MLMT) improved the resolution of *Leishmania* phylogenies and enabled population genetic analyses (*Gouzelou et al., 2012*; *Herrera et al., 2017*; *Kuhls et al., 2007*; *Schönian et al., 2011*) but are hard to compare when using different marker sets (*Schönian et al., 2011*). In contrast, genome-wide polymorphism data offers much greater resolution (*Downing et al., 2011*; *Rogers et al., 2014*), provides richer information on aneuploidy and other classes of variants, that is SNPs, small indels and structural variants, and enables insights into gene function from genome-wide studies of selection and association mapping (*Carnielli et al., 2018*; *Downing et al., 2011*). Moreover, advances in DNA sequencing technology together with the availability of reference genome assemblies for most of the clinically important species (*Downing et al., 2011*; *González-de la Fuente et al., 2019*; *Peacock et al., 2007*; *Real et al., 2013*; *Rogers et al., 2011*) in public databases (*Aslett et al., 2010*) now make it feasible to sequence collections of isolates and determine genetic variants genome-wide. Several studies on the *L. donovani* complex have applied such an approach including foci in Nepal (16 isolates, *Downing et al., 2011*), Turkey (12 isolates, *Rogers et al., 2014*), the Indian subcontinent (204 isolates, *Imamura et al., 2016*), Ethiopia (41 isolates from 16 patients, *Zackay et al., 2018*) and Brazil (20 and 26 isolates, respectively, *Teixeira et al., 2017*; *Carnielli et al., 2018*). However, genomic studies to date have addressed genome-wide diversity in geographically restricted regions, leaving global genome diversity in the species complex unknown.

We present whole-genome sequence data from isolates of the *L. donovani* species complex across its global distribution. Our genome-wide SNP data revealed the broad population structure of the globally distributed samples from the species complex. *L. infantum* samples from across the sampling range fall mainly into a single clade, while *L. donovani* is much more diverse, largely reflecting the geographical distribution of the parasites. As expected, parasites from the New World appeared closely related to parasites found in Mediterranean Europe. In addition to SNP diversity, we identified characteristic aneuploidy patterns of in vitro isolates shared across populations, variable heterozygosity between groups, differing levels of within-group linkage suggesting different recombination histories within geographical groups, and extensive structural diversity. This analysis reveals a much greater genetic diversity than suggested by previous, geographically-focused whole-genome studies in *Leishmania* and sets the scene for a new understanding of evolution in the *Leishmania donovani* species complex.

## Results

### Whole-genome variation data of 151 isolates of the *L. donovani* complex

We generated paired-end Illumina whole-genome sequence data from promastigote cultures of 97 isolates from the *L. donovani* complex. These sequence data resulted in a median haploid genome coverage ranging between 10 and 88 (median = 27) when mapped against the reference genome assembly of *L. infantum* JPCM5 (MCAN/ES/98/LLM-724; *Peacock et al., 2007*). These data were combined with subsets of previously published sequence data of strains of the *L. donovani* complex to represent previously sampled genetic as well as geographic diversity including parasites from Turkey (N = 11, *Rogers et al., 2014*), Sri Lanka (N = 2, *Zhang et al., 2014*), Spain (N = 1, *Peacock et al., 2007*), Ethiopia (N = 1, *Rogers et al., 2011*); N = 6, *Zackay et al., 2018*) and a subset of the extensive dataset available from the Indian subcontinent (N = 33, *Imamura et al., 2016*) resulting in a total of 151 isolates (*Supplementary file 1*, visualised at https://microreact.org/project/_FWlYSTGf; *Argimón et al., 2016*).

Accurate SNP variants were identified across 87.8% of the reference genome with a genotype quality of at least 10 (median = 99), indicating a < 0.1 (median = ~$10^{-10}$) probability of an incorrect genotype call. The remaining 12.2% could not be assayed as short reads could not be uniquely mapped to repetitive parts of the genome. This identified a total of 395,624 SNP sites out of the 32 Mb reference assembly. We also used these sequence data to infer extensive gene copy-number variation (91.5% of genes varied in dosage; 7,625/8,330 genes) and larger genome structure

variation, including copy numbers of individual chromosomes (aneuploidy) that is common in *Leishmania*. Together, these data represent the most comprehensive, global database of genetic variation available for any *Leishmania* species.

## Evolution of the *L. donovani* complex

Phylogenetic reconstruction based on whole-genome SNP variation clearly separated *L. infantum* from *L. donovani* strains. *L. donovani* separated into five major groups that coincide with geographic origin (*Figure 1A–B*, *Figure 1—figure supplement 1*) and show a strong signal of isolation-by-distance (IBD) between countries (0.76, p-value<=0.0001, Mantel test, *Supplementary file 2*). While the inferred root of the phylogeny is between *L. infantum* and *L. donovani*, groups within *L. donovani* showed similar levels of divergence as between the two species, with the deepest branches within *L. donovani* in East Africa. The largest *L. donovani* group in our collection, Ldon1, included samples from the Indian subcontinent, and could be further divided into two subgroups that separate samples from India, Nepal and Bangladesh from three samples of Sri Lankan origin; both subgroups displayed strikingly little diversity. The large number of isolates in Ldon1 is due to the extensive previous genomic work in this population (*Downing et al., 2011*; *Imamura et al., 2016*), which identified this as the 'core group' of strains circulating in the Indian subcontinent. The genetically and geographically closest group, Ldon2, was restricted to the Nepali highlands and also includes the more divergent sample, BPK512A1 (Ldon2 is the ISC1 group of *Imamura et al., 2016*). The latter isolate shared sequence similarity with a far more diverse group, Ldon4, of parasites from the Middle East (Iraq and Saudi Arabia) and Ethiopia (*Figure 1A*). Admixture analysis identified three additional samples (from Sudan and Israel), to be of mixed origin between groups Ldon3 and Ldon4. The Ldon3 group is restricted to Sudan and northern Ethiopia and an outlier sampled in Malta likely represents an imported case. Group Ldon5 displayed little diversity and is mainly confined to Southern Ethiopia and Kenya, with the rift valley in Ethiopia presumably restricting genetic exchange with Ldon3 through different sandfly vectors (*Gebre-Michael et al., 2010*; *Gebre-Michael and Lane, 1996*). A single outlier from this group, LRC-L51p, was sampled in India and again presumably represents an imported case of African origin.

In contrast, most of the samples of *L. infantum* clustered into a single group, Linf1, with relatively little diversity but a broad geographical distribution including Central Asia, the Mediterranean Region and Latin America but also very distinct lineages from the Western Mediterranean (*Figure 1A–B*). Admixture analysis using different numbers of total populations (*K*) divided the Linf1 group into two to three subgroups, separating samples from China, Uzbekistan and a single Israeli isolate, from two groups that both include samples from the Mediterranean region and Central/South America. This latter two subgroups correspond to MON-1 (31 samples of the largest subgroup) and non-MON-1 zymodemes (six samples from Europe, Turkey and Panama; *Figure 1A*, *Figure 1—figure supplement 1*) categorised by Multilocus Enzyme Electrophoresis (MLEE) (*Rioux et al., 1990*). Therefore, geography is not the main driver of parasite diversity across *L. infantum* in general nor within the globally distributed Linf1 group. This is also mirrored by only marginal isolation-by-distance correlations within Linf1 (0.20, p-value<0.05, Mantel test, *Figure 1—figure supplement 2A*, *Supplementary file 2*). However, IBD relationships are present within the 'MON-1' subgroup of Linf1 (0.47, p-value<=0.0001, Mantel test) and very pronounced between the non-American 'MON-1' strains (0.81, p-value<=0.0001, Mantel test, *Figure 1—figure supplement 2A*, *Supplementary file 2*). All 5 'MON-1' American samples formed a monophyletic sub-clade that was most closely related to parasite strains from Portugal, Spain, Italy and a single isolate from Israel suggesting a South-Western European origin of Central and South American *L. infantum* (*Figure 1—figure supplement 2A*). This result was still valid when including another 26 *L. infantum* isolates sampled from three states in Brazil (*Carnielli et al., 2018*). They all clustered in a single 'American' clade with little genetic diversity (*Figure 1—figure supplement 2B*). *L. infantum* in Central and Southern America, however, is not generally monophyletic as also one non-MON-1 *L. infantum* isolated in Panama was present in our dataset (*Figure 1—figure supplement 2*). For the zoonotic parasite *L. infantum*, 12 of our 30 MON-1 strains were isolated from dogs – previously also described as the prevalent zymodeme in dogs (*Pratlong et al., 2004*). For most countries this included isolates from human and non-human hosts, while samples generally clustered by geography (*Figure 1—figure supplement 2A*). This supports previous knowledge of dogs as a reservoir for human infection (*Alvar et al., 2004*).

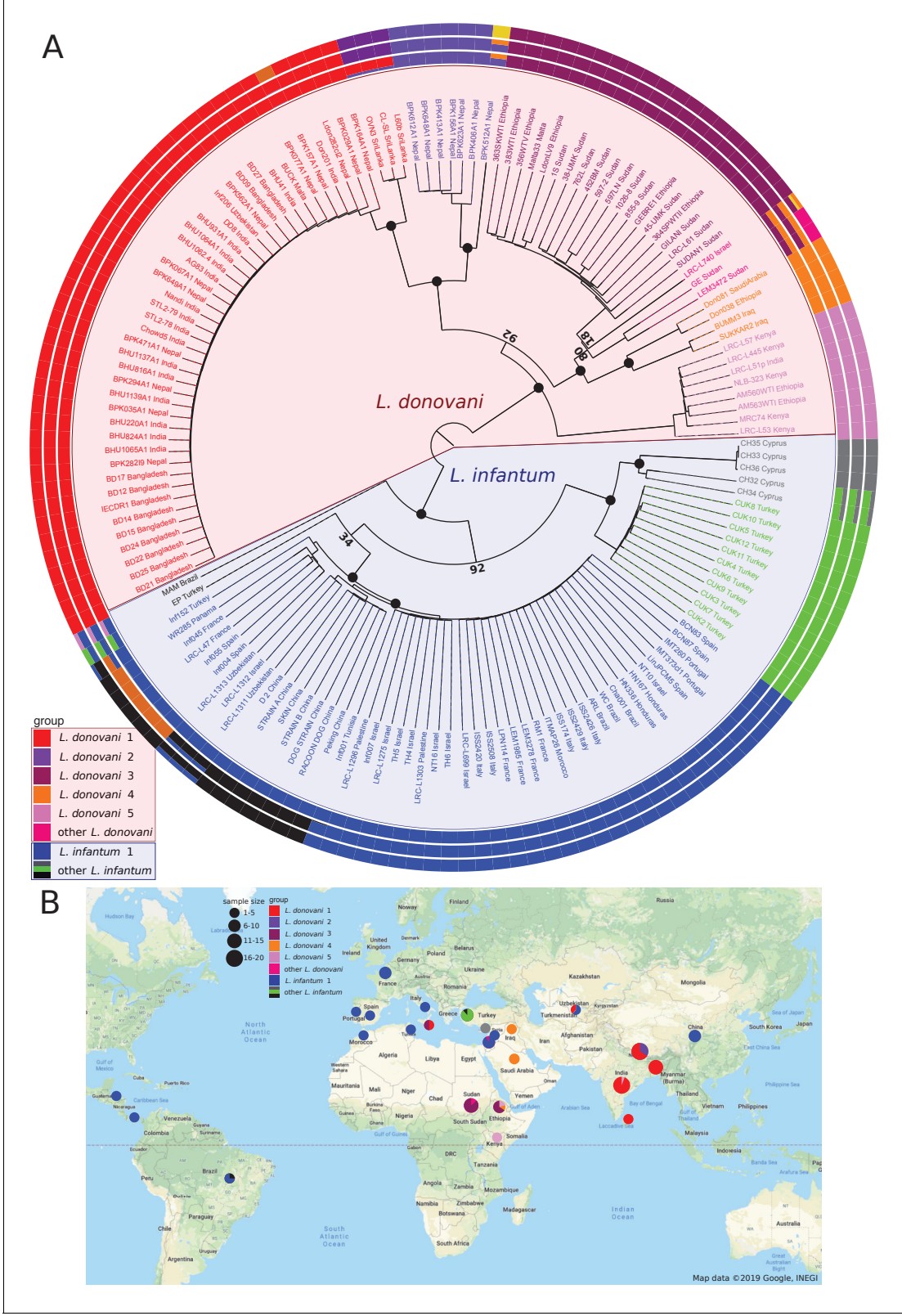

**Figure 1.** Sample phylogeny and distribution. (A) Phylogeny of all 151 samples of the *L. donovani* complex. The phylogeny was calculated with neighbour joining based on Nei's distances using genome-wide SNPs and rooted based on the inclusion of isolates of *L. mexicana* (U1103.v1), *L. tropica* (P283) and *L. major* (LmjFried) (outgroups not shown in the phylogeny). Bootstrap support is shown for prominent nodes in the phylogeny as black circles for values of 100% and otherwise the respective support value in % based on 1000 replicates. The groupings shown in the outer circles

*Figure 1 continued on next page*

*Figure 1 continued*
were calculated by admixture with $K = 8$, $K = 11$ and $K = 13$ (see Materials and methods). Groups labelled with different colours were defined based on the phylogeny and include monophyletic groups as well as groups that are polyphyletic and/or largely influenced by hybridisation (indicated by 'other'). (B) Map of the sampling locations. Groups are indicated by the different colours. Sample sizes by country of origin are visualised by the sizes of the circles.

The online version of this article includes the following figure supplement(s) for figure 1:

**Figure supplement 1.** Phylogenetic reconstruction of all 151 samples of the *L. donovani* complex.

**Figure supplement 2.** Sample phylogeny of the Linf1 group.

In contrast to the low diversity across the wide geographical range of the core *L. infantum* group, Linf1, the remaining samples of *L. infantum*, from Cyprus and Çukurova in Turkey, are genetically more distinct and showed unusual positioning in the phylogeny close to the split between *L. infantum* and *L. donovani*. Samples from the Çukurova region of Turkey (CUK, green) are considered to be a lineage descended from a single crossing event of a strain related to the *L. infantum* reference strain JPCM5 and an unknown *L. infantum* or *L. donovani* strain (*Rogers et al., 2014*). Isolates from Cyprus (CH, grey) are also divergent from the *L. infantum* group: these parasites were identified as *L. donovani* using MLEE, but the associated pattern of markers (MON-37) has been shown to be paraphyletic (*Alam et al., 2009*), so its species identity might be debateable. Our data suggest that the two slightly different Cypriot isolates (CH32 and CH34) are admixed between the Çukurova and remaining Cypriot strains. Two more isolates (MAM and EP; from Brazil and Turkey) are both highly divergent from any other isolates in the phylogeny, and appeared to be admixed between the Linf1 group and other lineages. As expected from the relatively high divergence of the CUK and Cypriot clades that have their origin from the centre of the sampling range, there is no overall IBD relationship across all *L. infantum* samples ($-0.12$, ns., Mantel test, *Supplementary file 2*). This suggests that in contrast to *L. donovani*, the majority of *L. infantum* shows little diversity, but diverse strains can co-localise in the case of non-MON-1 strains (see also *Guerbouj et al., 2001*) and can have diversified by hybridisation in case of the CUK strains.

## Aneuploidy

We observed extensive variation in chromosome copy number in our isolated strains in vitro, inferred from read coverage depth, with the pattern of variation being incongruent with the genome-wide phylogeny (*Figure 2—figure supplement 1*). Aneuploidy patterns are known to vary over very short time scales, even within strains and upon changing environments (*Sterkers et al., 2011*; *Dumetz et al., 2017*; *Lachaud et al., 2014*), although consistent patterns of aneuploidy have been observed within small groups of closely related cultured field isolates (*Imamura et al., 2016*). We took advantage of the greater diversity and global scope of our data to investigate somy patterns of cultivated promastigotes for individual chromosomes across geographically distinct groups. As expected, the majority of chromosomes had a median somy of two across isolates, apart from chromosomes 8, 9 and 23 and chromosome 31 with a median somy of three and four, respectively (*Figure 2A,C*, *Figure 2—figure supplement 2A*). However, trisomy was widespread with all chromosomes being overall trisomic in at least two isolates (2%) and at least half of all chromosomes were trisomic in $\geq$ 28 isolates (19%). In contrast, monosomy was rare – with only four chromosomes having somy of one in a single isolate each. As previously reported for *Leishmania* (e.g. *Akopyants et al., 2009*; *Downing et al., 2011*; *Imamura et al., 2016* ), chromosome 31 was unusual in being dominantly tetrasomic (81% of samples) and we observed no somy levels below three. Much of this pattern – general disomy, with occasional trisomy and sporadic higher dosage for most chromosomes – was consistent across the four largest groups, as was the high dosage of chromosome 31 (*Figure 2—figure supplement 2B*). Similarly, chromosome 23 showed a tendency to trisomy in all four groups, and chromosomes 8 and 9 were dominantly trisomic in three of the groups.

As some chromosomes appeared to be more frequently present at high copy numbers in our isolates, we investigated whether their copy numbers were also more variable. Copy number variability for each chromosome was estimated by the standard deviation (sd) in somy and was positively correlated between the four largest groups (*Figure 2B*). Correlations were much higher between three groups from diverse sampling locations, while correlations to the CUK group sampled in the

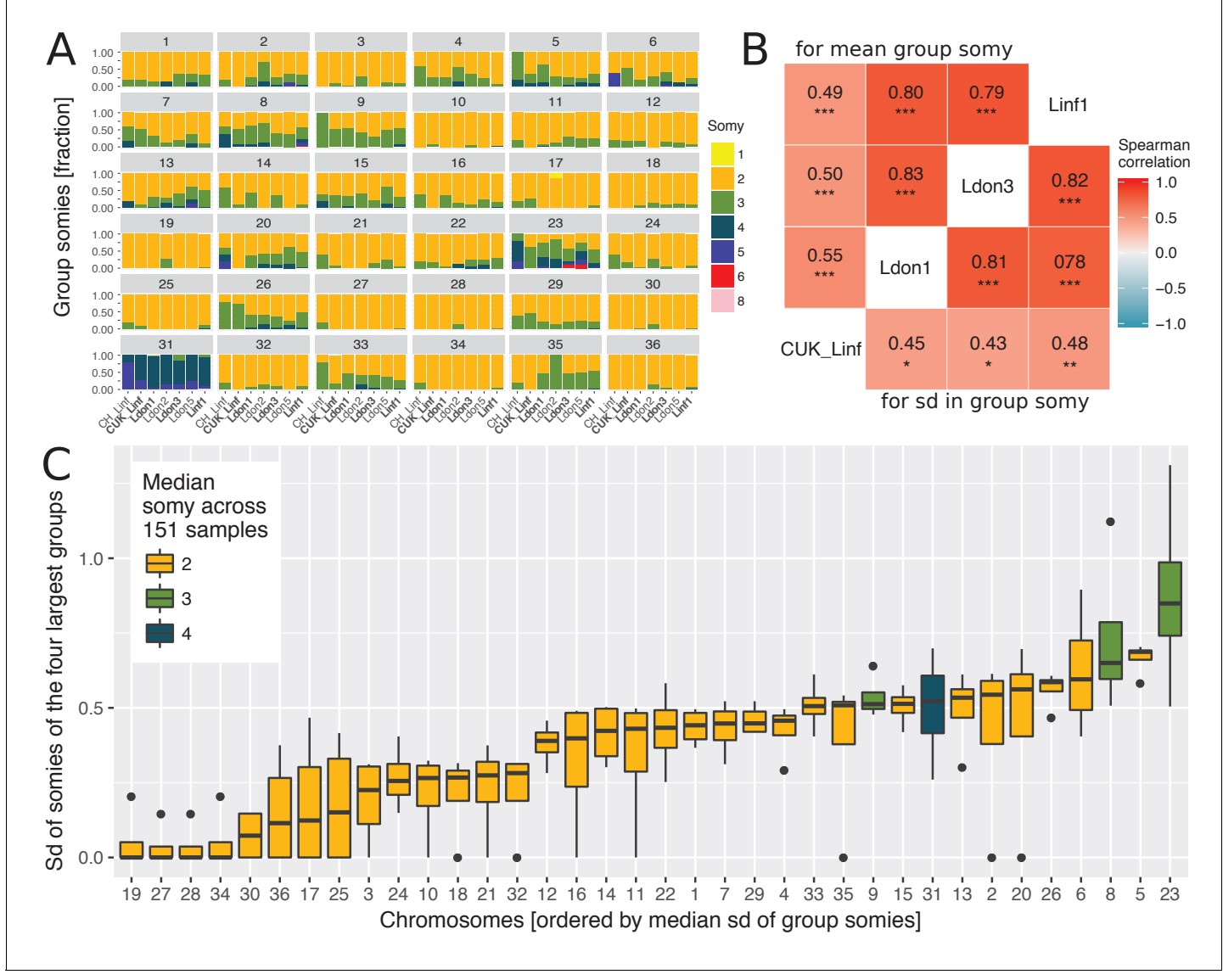

**Figure 2.** Chromosome-specific somy variability. (**A**) Somy variability is displayed for the 7 largest groups ($\geq$ 5 isolates) for each chromosome as fractions of isolates with the respective somies. The four largest groups ($\geq$ 9 samples per group) are indicated in bold. (**B**) The heatmap shows the Spearman correlations of chromosome-specific somy statistics between the four largest groups, measured as the mean group somies (upper triangle) and the standard deviation (sd) of chromosome somies (lower triangle), respectively. False discovery rates (FDR) of each correlation are indicated by asterisks (*: < 0.05, **: < 0.01, ***: < 0.001). (**C**) Boxplots show the distribution of variability in chromosome-specific somy across the four largest groups used as independent replicates across the species range. Medians estimate the chromosome-specific variation in somy.

The online version of this article includes the following figure supplement(s) for figure 2:

**Figure supplement 1.** Aneuploidy patterns across all 151 samples.
**Figure supplement 2.** Aneuploidy distributions for the different chromosomes.

Çukurova province were lower, suggesting a distinct pattern of aneuploidy variability in this group – perhaps due to its hybrid origin (*Rogers et al., 2014*). Given the positive correlations between independent groups, we investigated chromosome-specific variation in somy using the four independent groups (*Figure 2C*). A few chromosomes including 19, 27, 28 and 34 showed almost no variation, while several chromosomes showed very high variation in chromosome copy number with the top five chromosomes being 23, 5, 8, 6 and 26 (*Figure 2C*). This indicated that some chromosomes have higher propensities for chromosome aneuploidy turnover than others.

## Heterozygosity

Samples varied greatly in genome-wide heterozygosity: 70% of the isolates in our collection showed extremely low heterozygosity (<0.004; see Materials and methods) corresponding to between 23 and 2057 (median = 80) heterozygous sites per sample. The remaining high-heterozygosity samples largely showed heterozygosities up to ~0.02 (equivalent to 15,281 heterozygous sites per sample) with a few outliers exceeding this threshold and reaching a heterozygosity of 0.065 in one isolate (MAM, 50,543 heterozygous sites) (*Figure 3A*). For almost all isolates the majority of genome-wide 10 kb windows had almost no heterozygous sites: only 11 isolates had a median count of heterozygous sites per window greater than zero (*Figure 3—figure supplement 1*). This predominant homozygosity for the majority of isolates of the *L. donovani* complex was in striking contrast to expectations for sexual populations under Hardy-Weinberg equilibrium, or for clonally reproducing populations: clonal reproduction is expected to increase heterozygosity, as single mutations cannot be assorted to form novel homozygous genotypes (*Balloux et al., 2003*; *De Meeûs et al., 2006*; *Weir et al., 2016*). Most main groups were dominated by samples of low heterozygosity, with the exception of the Ldon3 group and the CUK group of hybrid *L. infantum* isolates (*Rogers et al., 2014*). Other high-heterozygosity isolates mainly appeared in positions intermediate between large groups in the phylogeny, and showed mixed ancestry in the admixture analysis (e.g. isolates MAM, EP, CH32, CH34, GE, LEM3472, LRC-L740; *Figure 1A*), leading us to hypothesise that they represent recent hybrids between the distinct, well-differentiated populations.

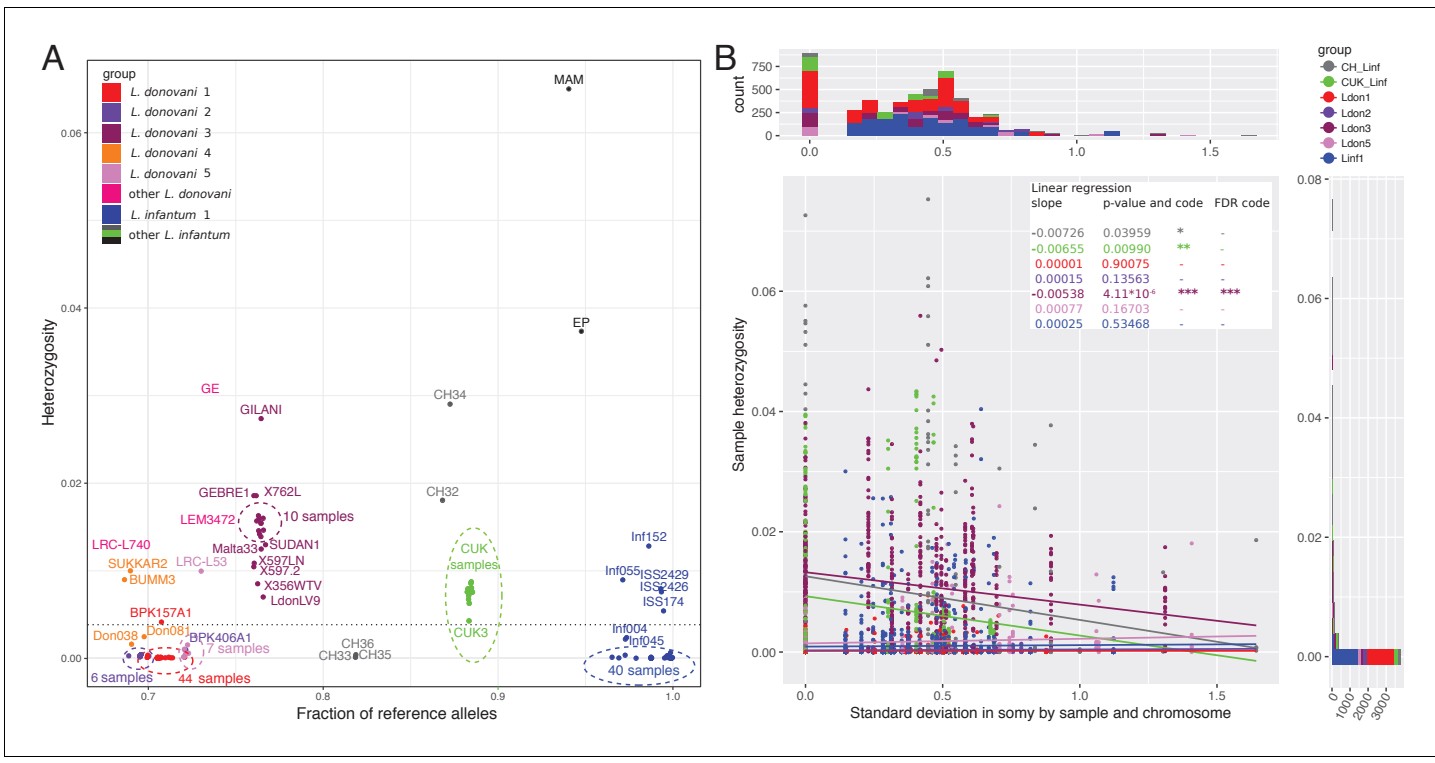

**Figure 3.** Whole genome sample heterozygosities. (**A**) Whole genome heterozygosities versus fraction of reference alleles. The fraction of reference alleles is calculated across all 395,602 SNP loci in the data set. Isolate names are written unless they are present in dense clusters indicated by dashed-line circles. Groups are indicated by colour as defined in figure 1. The dashed horizontal line at a genome-wide heterozygosity of 0.004 was chosen to separate samples with putative recent between-strain hybridisation history. (**B**) Relationship between chromosome-specific somy variability and sample heterozygosity. The scatterplot describes the relationship between the standard deviation in chromosome-specific somy by group (groups with ≥ 5 samples) against the chromosome-specific sample heterozygosity. Linear regressions were performed for each group. Asterisks indicate statistical significance of the estimated regression slope with *: < 0.05, **: < 0.01, ***: < 0.001 or '-' for not significant. Marginal histograms on the top and on the right correspond to the x-values and the y-values of the scatterplot, respectively. Groups are indicated by the different colours.

The online version of this article includes the following figure supplement(s) for figure 3:

**Figure supplement 1.** Distribution of heterozygous sites across the genome.

The low heterozygosity together with strong genetic signatures of inbreeding in *Leishmania* had previously been identified using MLST and microsatellite data, and has generally been attributed to extensive selfing between cells from the same clone (*Ramírez and Llewellyn, 2014*; *Rougeron et al., 2009*). However, an alternative explanation could be that frequent aneuploidy turnover also reduces within-cell heterozygosity if an alternate haplotype is lost during somy reduction (*Sterkers et al., 2014*). We therefore tested whether the chromosome-specific variation in somy for each group was negatively correlated with chromosome-specific sample heterozygosity, as a high turnover rate could reduce within-strain heterozygosity. Linear regressions for the different groups showed negative slopes for three of seven groups but only the slope for the Ldon3 group was significant after multiple testing correction (*Figure 3B*). For the four groups, Ldon1, Ldon2, Ldon5 and Linf1, where the regression slope was almost zero, the chromosomes were almost completely homozygous which might make potential effects undetectable (*Figure 3A,B*). The data for the remaining groups is in accordance with a reduction in heterozygosity with aneuploidy turnover. However, to establish presence and effect sizes of a reduction in heterozygosity due to aneuploidy turnover direct experiments and more accurate estimates of aneuploidy turnover are needed, particularly using in vivo parasites.

## Genomic signatures of hybridisation

To clarify the relationship between the high heterozygosity of some isolates, their phylogenetic position and the signatures of admixture, we examined the genomes of all 46 isolates with genome-wide heterozygosity greater than 0.004 in more detail for signs of past hybridisation (*Figure 3A*, row A1 in *Table 1*). This threshold was chosen to include the majority of samples that had putative hybrid ancestry in the admixture analysis, including the Çukurova samples of known hybrid origin (*Rogers et al., 2014*). The few isolates with lower heterozygosity but other evidence of admixture were also investigated (BPK512A1, L60b, CL-SL and OVN3 between groups, and LRC-L1311, LRC-L1312 and LRC-L1313 between subgroups; rows A2 and B6 in *Table 1*), but identifying details beyond admixture results was difficult with only a few SNPs available (e.g. *Figure 4—figure supplements 1A* and *2D*). For the 46 high-heterozygosity isolates (*Table 1*), we inspected the distribution of heterozygous sites along each genome, looked for blocks of co-inherited variants and investigated patterns of allele-specific read coverage (i.e. sample allele frequency) across each chromosome. We also inferred maxicircle kinetoplast (mitochondrial) genome sequences: as kDNA is considered to be uniparentally inherited (*Akopyants et al., 2009*; *Inbar et al., 2013*), the phylogeny for these sequences should identify one parent of any hybrid isolates.

28 of the 46 high heterozygosity isolates appeared to represent genuine hybrid lineages (rows B1, B2 and B4 in *Table 1*), and for 17 of these, likely parents could be assigned (row B2 in *Table 1*). The largest group with identified parents is the Turkish isolates from Çukurova province (*Rogers et al., 2014*). Additionally, two Cypriot isolates (CH32 and CH34) showed patches of homozygosity closely related to the other Cypriot isolates and the Turkish CUK hybrids (*Figure 4*, *Figure 4—figure supplement 1A*). Therefore, CH32 and CH34 likely represent hybrids closely related to the CUK hybrids, but clearly derived from an independent hybridisation event to the CUK population itself (*Figure 1A*). Another Turkish isolate (EP) appeared to have a similar evolutionary history with putative parental strains from the Linf1 and the CUK hybrids (*Figure 4*). In contrast to previous hybrids, for EP, there were entire homozygous chromosomes that resembled either of the two putative parental groups (chromosomes 4, 12, 22 and 32 for one and 11, 23 and 24 for the other parent; *Figure 4*). Phylogenetic analysis of the kDNA maxicircles further showed identical sequences to the Cypriot hybrid samples (CH23 and CH34, *Figure 4—figure supplements 3*, *4*, *Supplementary file 3*). Additionally, on two chromosomes, 5 and 31, allele frequency distributions in the EP isolate were not compatible with a single, clonal population of cells suggesting the presence of a second but very closely related low frequency clone in this sample (*Figure 4—figure supplements 2*, *5*). We also saw discrete patches of heterozygous and homozygous variants in two isolates from East Africa (GE and LEM3472) and one from Israel (LRC-L740) that did not fit into any of the main *L. donovani* groups. These isolates appeared admixed between the North Ethiopia/Sudan group (Ldon3) and the *L. donovani* group present in the Middle East (Ldon4) (*Figures 1A* and *4*, *Figure 4—figure supplement 1A*). For sample GE, kDNA further confirmed that one putative parent came from the Ldon3 group (*Figure 4—figure supplement 3*). All the isolates from the Ldon3 group, were also highly

**Table 1.** Summary of the hybrid analysis.

| Category | ID | Description | Interpretation | # Samples | Fraction of samples | Sample identities |
|---|---|---|---|---|---|---|
| Initial definition of the 53 (35%) putative hybrids | A1 | 'High' genome-wide heterozygosity (>=0.004) | initial indicator for putative hybrids | 46 | 30% | BPK157A1, BUMM3, CH32, CH34, CUK10, CUK11, CUK12, CUK2, CUK3, CUK4, CUK5, CUK6, CUK7, CUK8, CUK9, EP, GE, GEBRE1, GILANI, Inf055, Inf152, ISS174, ISS2426, ISS2429, LdonLV9, LEM3472, LRC-L53, LRC-L61, LRC-L740, Malta33, MAM, SUDAN1, SUKKAR2, 1026–8, 1S, 356WTV, 363SKWTI, 364SPWTII, 38-UMK, 383WTI, 45-UMK, 452BM, 597–2, 597LN, 762L, 855–9 |
| | A2 | 'Admixed' between groups (admixture analysis) | initial indicator for putative hybrids | 15 | 10% | BPK512A1, CH32, CH34, CL-SL, EP, GE, Inf152, L60b, LEM3472, LRC-L1311, LRC-L1312, LRC-L1313, LRC-L740, MAM, OVN3 |
| Detailed investigation of the 53 (35%) putative hybrids | B1 | Heterozygous sites distributed relatively evenly across the genome and allele frequency profiles match coverage based somy estimates | putative patterns of sexual crossing (F1/F2+), however, cannot be verified without identified putative parents; alternative explanation could be new mutations that are dominating the sample population through a recent bottleneck (e.g. cloning) | 18 | 12% | Inf055, GEBRE1, LdonLV9, LRC.L61, SUDAN1, 1026–8, 1S, 356WTV, 363SKWTI, 364SPWTII, 38-UMK, 383WTI, 45-UMK, 452BM, 597–2, 597LN, 762L, 855–9 |
| | B2 | Evidence for parents between different groups (or between two distinct strains as previously shown for the CUK samples) alternating in the genome in a block like pattern | putative patterns of sexual crossing (F2+), that is 'hybrids' | 16 (+1) | 10% (11%) | CH32, CH34, CUK10, CUK11, CUK12, CUK2, CUK3, CUK4, CUK5, CUK6, CUK7, CUK8, CUK9, EP, GE, LEM3472, (LRC-L740) |
| | B3 | Extreme allele frequency variants only | mixture of two different high versus low frequency clones or low frequency new mutations distributed across haplotypes in the sample | 7 | 5% | BPK157A1, Inf152, ISS174, ISS2426, ISS2429, LRC-L53, MAM |
| | B4 | Intermediate peak allele frequency distributions including extreme frequency peaks | mixture of scenarios B1 and B3, that is as B3 but high frequency clone has heterozygous sites itself | 4 | 3% | BUMM3, LRC-L740, Malta33, SUKKAR2 |
| | B5 | no clear peak pattern of allele frequencies (several peaks at atypical frequencies) | mixture of several clones | 1 | 0.01% | GILANI |
| | B6 | to few heterozygous sites present to draw further conclusions beyond admixture results | signatures are shadowed by too little segregating variation | 7 | 5% | BPK512A1, CL-SL, L60b, LRC-L1311, LRC-L1312, LRC-L1313, OVN3 |

heterozygous and so potentially hybrids, but we cannot exclude other possible origins for this heterozygosity (*Figures 3A* and *4* , *Figure 4—figure supplement 1*, *Table 1*).

While the CUK samples are known to be of hybrid origin between a JPCM5-like *L. infantum* isolate and an unidentified parasite from the *L. donovani* complex (*Rogers et al., 2014*), our admixture results did not suggest hybridisation between genetic groups present in our dataset. This still held when varying *K* (the specified number of subpopulations) from 2 to 25 (*Figure 4—figure supplement 6*). We therefore took a haplotype-based approach to increase the power to identify putative parents of these hybrids similar to that in *Rogers et al. (2014)*, but now compared them to our larger set of isolates. We identified the largest homozygous regions in the CUK genomes: that is those that were either almost devoid of SNP differences to the JPCM5 reference genome or those that had a high density of fixed differences but lacked heterozygous sites, and generated phylogenies for these regions (*Figure 4—figure supplement 7*; see Materials and methods). Trees for the four largest regions (155 kb – 215 kb) placed the JPCM5-like parent close to *L. infantum* samples from China, rather than to the classical MON-1 and non-MON-1 Mediterranean subgroups

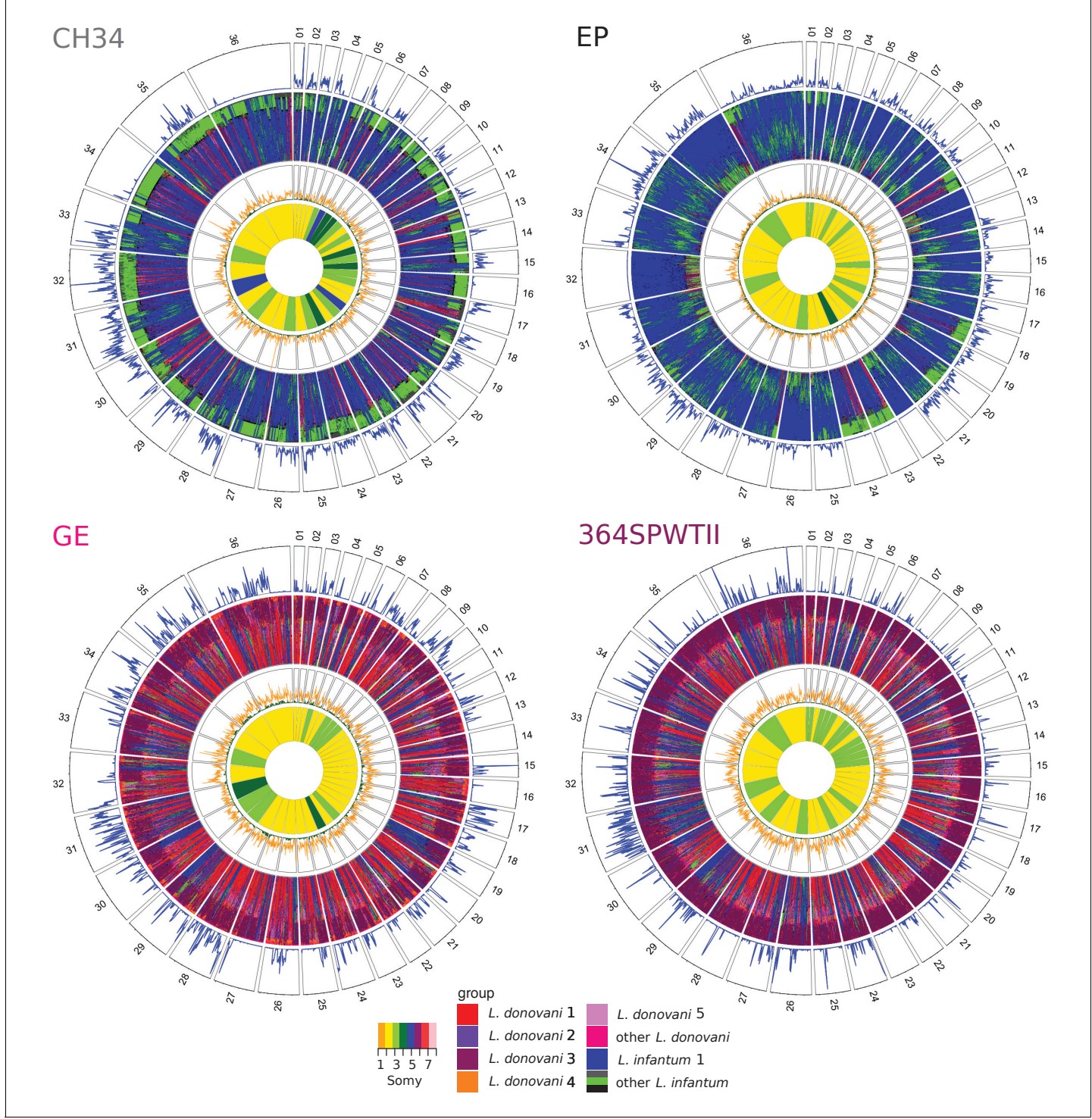

**Figure 4.** Window-based analysis of relatedness. Each circos plot shows four different genomic features of the isolate named in each top left corner. In the four different rings, pies correspond to the different chromosomes labelled by chromosome number. The three outer rings show a window-based analysis for a window size of 10 kb. Starting from the outer ring, they show: 1. Heterozygosity with the number of heterozygous sites ranging from 0 to 98, 146, 90 and 85 sites per window for CH34, EP, GE and 364SPWTII, respectively, 2. A heatmap coloured by groups of the 60 genetically closest isolates based on Nei's D and starting with the closest sample at the outer margin and the 60th furthest isolate at the inner margin, 3. Nei's D to the closest (green) and the 60th closest isolate (orange) scaled from 0 to 1. The innermost circle shows the colour-coded somy.

The online version of this article includes the following figure supplement(s) for figure 4:

**Figure supplement 1.** Window based analysis of relatedness for a subset of samples.

*Figure 4 continued on next page*

*Figure 4 continued*

**Figure supplement 2.** Allele frequency distributions by isolate.
**Figure supplement 3.** Phylogenetic tree based on the maxicircle DNA.
**Figure supplement 4.** Coverage of maxicircle DNA.
**Figure supplement 5.** Somy evaluation based on allele frequency profiles.
**Figure supplement 6.** Sample phylogeny and admixture analysis across a range of *K* values.
**Figure supplement 7.** Genomic regions used for haplotype-based parent identification.
**Figure supplement 8.** Putative parents of CUK samples.
**Figure supplement 9.** Sample phylogeny based on genomic SNP variation including phased samples with skewed allele frequency spectra.
**Figure supplement 10.** Heterozygosity of artificial F1 hybrids.
**Figure supplement 11.** Verification of skewed allele frequency spectra in a subset of isolated strains.
**Figure supplement 12.** Correlation between somies and heterozygosities across chromosomes.

(*Figure 4—figure supplement 8A*). Trees for the putative other parent always grouped CUK with CH samples similarly to the phylogeny of the maxicircle DNA (*Figure 4—figure supplement 3*), suggesting these as closest putative parents to the CUK group in our sample collection (*Figure 4—figure supplement 8B*). The phylogenetic origin of the CH samples, however, still remained uncertain: in these four phylogenies the CH samples clustered twice next to the Ldon4 group, once next to Linf1 and once between both species. A haplotype-based approach as used for the CUK samples, and polarizing on several different isolates also did not give clear results (data not shown).

## Isolates with genetically distinct (sub-)clones

Unexpectedly, for 12 of the remaining isolates (rows B3 – B5 in *Table 1*), many of the heterozygous sites were present at extreme (high/low) allele frequencies (11 isolates) or at multiple intermediate frequencies (isolate GILANI), incompatible with the allele frequencies expected based on chromosomal somy (*Figure 4—figure supplements 2*, *5*). We suspect that these isolates represent a mixture of multiple cell clones. However, as low frequency variants are more at risk of being false positive SNP calls, we additionally selected a subset of the highest confidence SNPs to verify the observed frequency patterns (see Materials and methods). The MAM isolate had the highest heterozygosity in our collection: it only had 178 homozygous differences to the JPCM5 reference, but 50,534 heterozygous sites, with a frequency of the reference allele of ~0.92 across all chromosomes (*Figure 4—figure supplement 2*). Phylogenies for inferred haplotypes of these low-frequency variants were closest but not part of the Ldon5 group (*Figure 4—figure supplement 9*), although this was somewhat variable between chromosomes (*Figure 4—figure supplement 9B–D*). We concluded that the MAM sample is most likely a mixture between a JPCM5-like *L. infantum* strain at high (~0.92) and an *L. donovani* related to Ldon5 at low (0.08) sample frequency. Due to the low frequency of the 2nd strain it might be that alleles have been missed for SNP calling and therefore the calculated sample heterozygosity is lower than expected for interspecies F1 crosses (see *Figure 4—figure supplement 10*). Similarly, the few heterozygous isolates within several *L. donovani* groups, BPK157A1 in Ldon1, Malta33 and GILANI in Ldon3, SUKKAR2 and BUMM3 in Ldon4 and LRC-L53 in Ldon5 (*Figure 3A*) all appeared to be mixtures of two clones from within the respective group (*Figure 4—figure supplement 9*) apart from GILANI, which might be a more complex mixture (*Figure 4—figure supplement 2*). For two of those samples the high number of within sample SNPs is due to segregating clones at high and low frequency (BPK157A1, LRC-L53 see row B3 in *Table 1*). For the other samples (BUMM3, Malta33, SUKKAR2; row B4 in *Table 1*) the majority of SNPs come from heterozygous sites of a putative hybrid with a smaller fraction of SNPs owing to an additional related low frequency clone (*Figure 4—figure supplement 2*). However, as one isolate from this subset (BPK157A1) was re-grown from a single cell prior to sequencing (*Supplementary file 1*), we cannot be sure that these variants are due to a mixture of clones. We ruled out false positive SNP calls by identifying 216 of the highest quality SNPs that show the extreme frequency pattern (*Figure 4—figure supplement 11*; Materials and methods), however, alternate explanations including incomplete cloning or changes during in vitro culture post-cloning also seem unlikely. Highly heterozygous isolates from *L. infantum* (ISS174, ISS2426, ISS2429 and Inf152 in Linf1) also had skewed allele frequency distributions *Figure 4—figure supplements 2*, *11*), and therefore likely represent either mixed clone isolates or samples that have evolved significant diversity during in vitro growth.

Samples, ISS174, ISS2426 and ISS2429, showed a strong positive correlation of chromosomal heterozygosity and somy not found in any other samples (*Figure 4—figure supplement 12*). We speculate that these isolates may have accumulated substantial numbers of new mutations most likely while maintaining relatively stable chromosome copy number during in vitro culture. Consequently, we expect relatively more mutations on chromosomes with a higher chromosome dosage, resulting in higher heterozygosity of high somy chromosomes.

## Population genomic characterisation of the groups

Sexual recombination is not obligate in the *Leishmania* lifecycle and appears to be rare in many natural populations (*Imamura et al., 2016*; *Ramírez and Llewellyn, 2014*; *Rougeron et al., 2009*). We thus examined patterns of linkage disequilibrium (LD) between *Leishmania* populations as a clue to the frequency of sexual recombination, bearing in mind that LD can be affected by underlying population structure. LD estimates further depend on the frequency of recombination, the population size, demographic history (*Slatkin, 1994*) and the size of sample taken from the population (see also *Figure 5A* versus *Figure 5—figure supplement 1*). We subsampled larger groups to identical group sizes and found strong differences between groups in LD decay with genomic distance (*Figure 5A*). Linkage was strongest in the Ldon2 group with mean LD estimates around 0.9 regardless of genomic distance between SNPs, even when comparing sites on different chromosomes. The *L. infantum* groups (Linf1 and the CUK samples) started with high mean LD values for 1 kb distances above 0.9 and 0.8, respectively, and dropped to ~0.5 for 100 kb distances and to ~0.4 and ~0.3 between chromosomes. Ldon3 and Ldon5 groups had the lowest LD estimates: at up to 1 kb distances LD had mean values of ~0.8 and 0.6 for Ldon3 and Ldon5, respectively, and dropped to ~0.2 for distances $\geq$ 50 kb in both groups and remained at those levels between chromosomes. All of these trends were relatively consistent among three independent subsamples from each of the larger groups, but the pattern was more complex for Ldon1. Here, the mean LD had a flat distribution with genomic distance like the Ldon2 group but at a much lower LD level, and showed significant variation between 3 subsamples (*Figure 5B*): two of the three subsamples showed low but very variable LD, and the third showed consistently high LD with distance. Low LD replicates were based on samples with a greatly reduced number of within-replicate SNPs (683 and 685 in R1 and R3 versus 23,303 SNPs in replicate R2). In the low LD replicates the majority of SNPs were singletons or present in only two copies, while in replicate R2 the majority of minor alleles were present at four copies (*Figure 5—figure supplement 2A*). Mean LD estimates across the entire Ldon1 group were also consistent at high levels above 0.8 independent of genomic distance (*Figure 5—figure supplement 1*). We conclude that the substructure described for samples from the Indian subcontinent (*Imamura et al., 2016*) is responsible for varying LD estimates of the subsamples, with low LD replicates due to sampling only closely-related subgroups that only differ in a small number of isolate-specific variants that are most parsimoniously described by recent mutations (*Figure 5B*). While the level of LD in a population cannot be used to directly quantify the frequency of recombination due to the contribution of demographic factors, we interpret a gradual decrease of LD with distance as a signal of frequent recombination occurring in those populations.

The groups also differed in their allele frequency distributions (i.e. the site frequency spectra, SFS). In a diploid, panmictic and sexually recombining population of constant population size neutral sites should segregate following a reciprocal function (*Ferretti et al., 2018*; *Wright, 1938*). While we would not predict *Leishmania* populations to exactly follow these expectations, most of the groups (Ldon1, Ldon2, Ldon5 and Linf1) were dominated by low frequency variants (*Figure 5—figure supplement 2*). In contrast, intermediate frequency variants were frequent in Ldon3 and even dominated variation in the *L. infantum* CUK samples. The CUK group had been suggested to have largely expanded clonally from a single hybridisation event between diverse strains with little subsequent hybridisation (*Rogers et al., 2014*). This scenario might explain why polymorphic sites, generated by the hybridisation of diverse strains and common to the majority of samples can be at intermediate population frequency. This group history also agrees with stronger LD over short distances due to shared blocks that may be broken up by rare subsequent hybridisation and recombination events. For the Ldon3 group increased intermediate frequency alleles combined with a strong decline of LD with distance might suggest that old variants are segregating in the group at high frequencies, due to relatively frequent hybridisation between clones within this group.

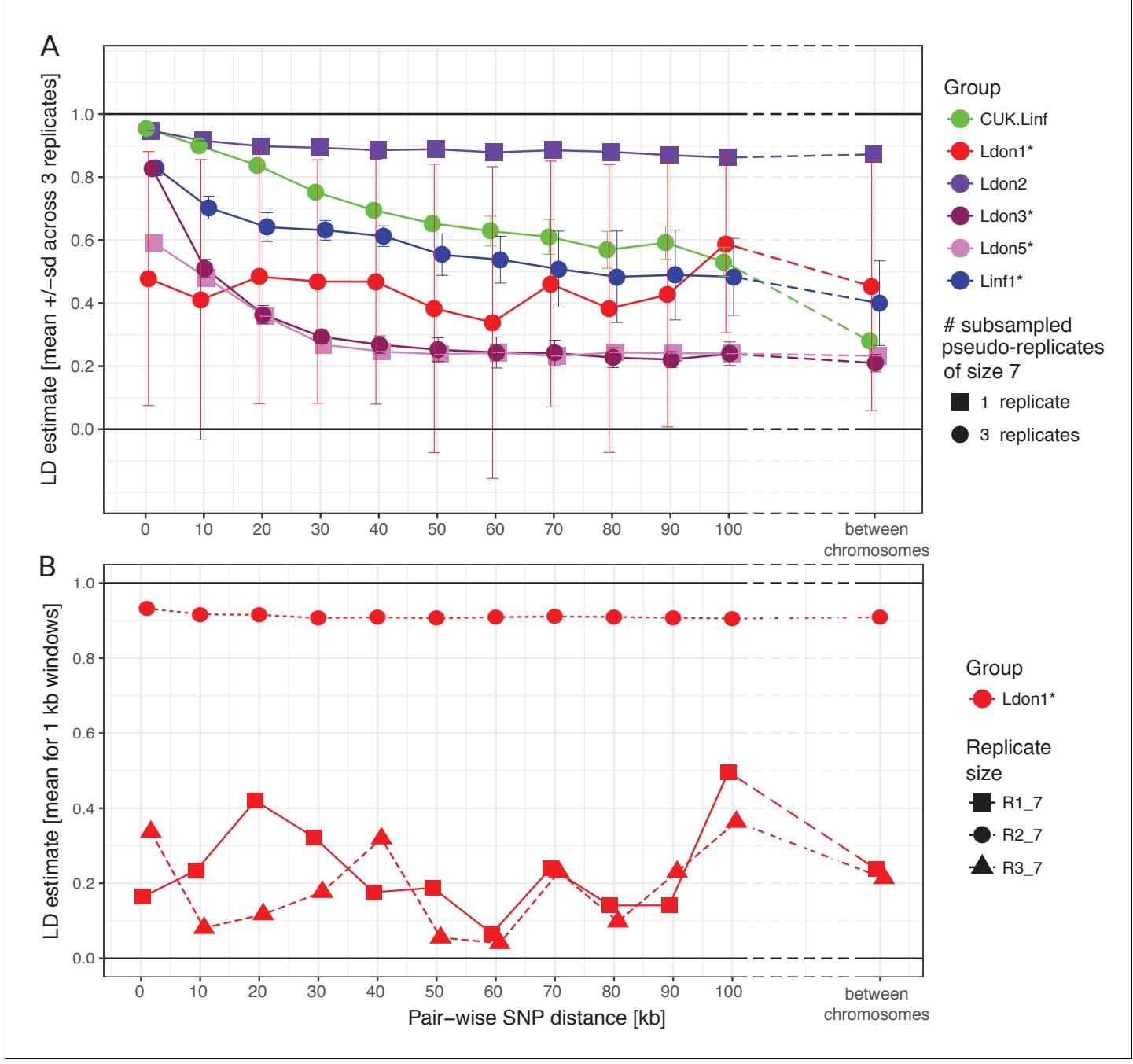

**Figure 5.** LD decay with genomic distance. (**A**) LD decay was measured for the six largest groups removing isolates that were identified as putative strain mixtures (indicated by *; see Materials and methods). Groups with more than seven isolates per group were sub-sampled to three pseudo-replicates of seven isolates (round symbols) to make LD estimates comparable between groups. Mean and standard deviation across the three pseudo-replicates are shown where applicable. Groups with only seven isolates were not sub-sampled and are indicated by squared symbols. (**B**) LD decay with distance is shown for the three pseudo-replicates for the Ldon1 group. (**A and B**) Data for individual replicates was calculated as means of 1 kb windows for SNP pairs of the stated genomic distance. For LD estimates between chromosomes, 100 SNPs were randomly sampled per chromosome and means across all pair-wise combinations between chromosomes are shown. This procedure was done twice independently but as differences between both such replicates were negligible, only the results of one replicate are shown.

The online version of this article includes the following figure supplement(s) for figure 5:

**Figure supplement 1.** LD decay with genomic distance.

**Figure supplement 2.** Folded site frequency spectra of the six largest groups.

To identify genomic differences between the major groups, we determined the fixation index ($F_{ST}$) for all SNP variants among pairs of groups, excluding samples identified as between group mixtures (*Table 1* B3 and B4) or hybrids between groups (*Table 1* B2, except CUK samples). Most SNP sites segregating within each pair of groups were found to be population-specific, that is $F_{ST} = 1$, in 10 out of 15 pairs (*Figure 6A*). This confirmed that most groups are well differentiated from each other with limited gene flow between them. This high level of differentiation allowed us to identify between 6,769 and 26,145 potentially differentially fixed 'marker' SNPs for each group (*Figure 6B*, *Supplementary file 4*). These markers can be useful in diagnosing parasite infections from particular

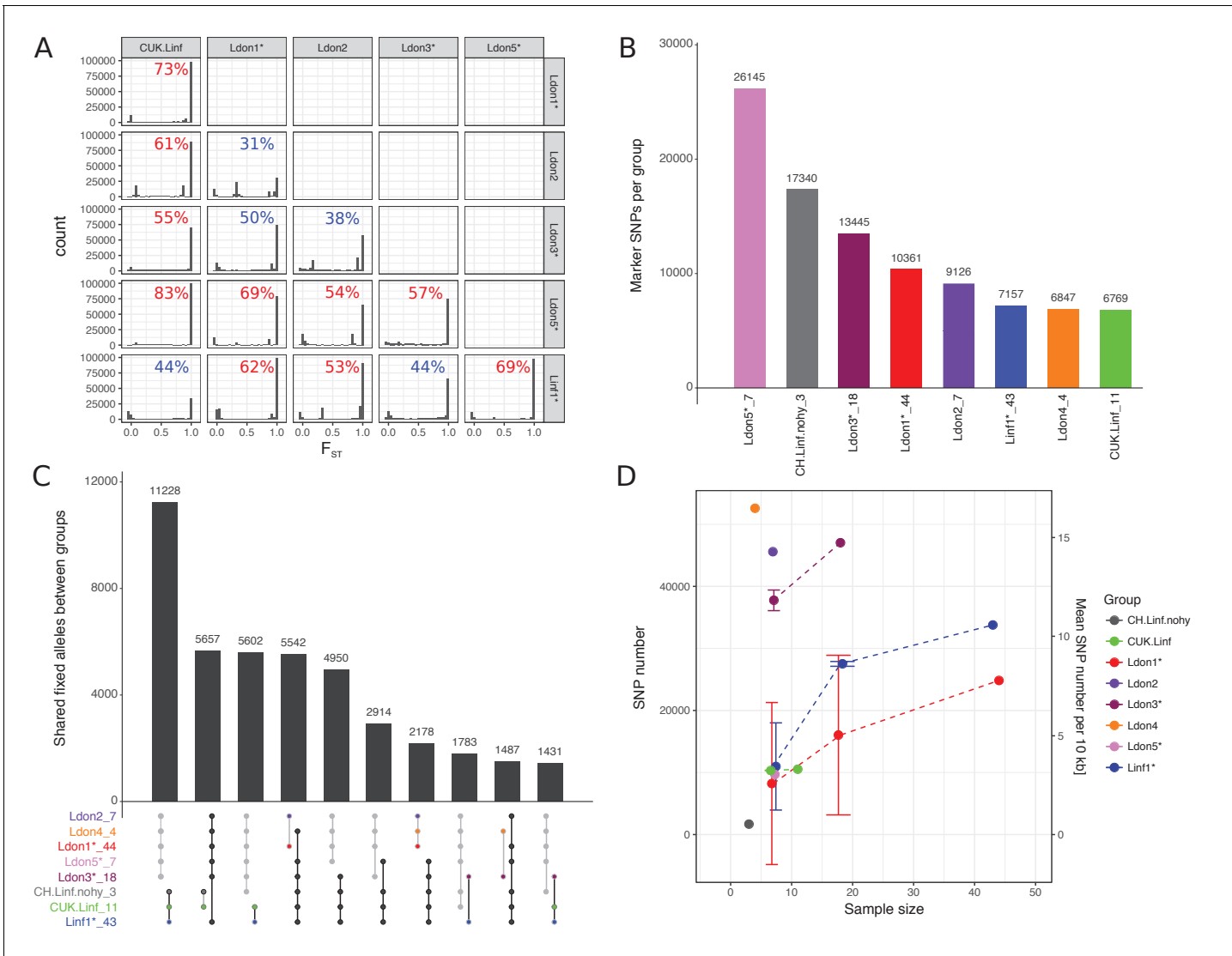

**Figure 6.** Differentiated and segregating SNPs between and within groups. For this analysis isolates that were shown to be mixtures of clones or hybrids between groups were removed (indicated by '*', see also Materials and methods). Groups sizes after removal of those isolates are specified in panels A and C. (**A**) $F_{ST}$ values between pairwise group comparisons. The fraction of differentially fixed SNPs ($F_{ST} = 1$) for each pairwise group comparison is indicated at the top right corner of each plot. Percentages larger than 50% are coloured in red, otherwise blue. (**B**) The number of marker SNPs for each group, that is SNPs that are differentially fixed in one group versus all others. (**C**) Number of SNPs that are differentially fixed between sets of groups. Groups fixed for the same allele are indicated in the bottom panel through connecting points corresponding to the specific groups. Grey and black lines connect sets of groups monomorphic for the alternate and reference allele, respectively. (**D**) Number and density of SNPs segregating in the respective groups. As sample sizes of the different groups vary, figures are also shown for three random sub-samples of the larger groups. Results of sub-sampling are displayed as mean and sd.

The online version of this article includes the following figure supplement(s) for figure 6:

**Figure supplement 1.** Polymorphism sharing between groups.

groups, but might not be fixed in populations identified based on a few isolates only. Despite this differentiation, many variants remained that were fixed in combinations of groups. Most of these SNPs supported the species split, between *L. infantum* and *L. donovani*, with 11,228 differentially fixed SNPs (*Figure 6C*). Within-group genetic diversity varied substantially between groups ranging from less than 1 SNP/10 kb within the three CH samples to ~16 SNPs/10 kb in Ldon4 (*Figure 6D*). Subsampled groups of seven isolates typically had ~3 SNPs/10 kb, while the two more polymorphic groups of *L. donovani* had SNP densities of ~12 and ~14 SNPs/10 kb. Most within-group segregating variation was group-specific: no SNPs segregated within all eight groups. The most widespread polymorphisms are 4 SNPs shared between 6 groups and 25 SNPs segregating in at least five of the eight groups and might be putative candidates for SNPs under balancing selection (*Figure 6—figure supplement 1*, *Supplementary file 5*).

## Copy number variation

To assess the importance of genome structure variation in *Leishmania* evolution, we identified all large sub-chromosome scale copy number variants (CNVs) within our isolates (duplications and deletions $\geq$ 25 kb; see Materials and methods). In total, 940 large CNVs were found, an average of ~6 per sample. 75% of these large variants had a length $\leq$ 40 kb and only ~3% were > 100 kb with the largest variant of 675 kb (*Supplementary file 6*, *Figure 7—figure supplement 1*). Most of these very large variants ( > 100 kb), were located on chromosome 35 (*Figure 7—figure supplement 2*). Interestingly, those were all either deletions or duplications close to the 3' and 5' end of the chromosome, respectively. All those duplications contained the previously described CD1/LD1 locus (*Figure 7—figure supplement 2*; *Sunkin et al., 2001*; *Kündig et al., 1999*; *Lemley et al., 1999*). In total, we found at least 9 different duplicated sequences spanning the CD1/LD1 locus, present in 13 of our 151 isolates (*Supplementary file 7*, *8*). The frequency of large CNVs varied among chromosomes but was not associated with chromosome length for duplications (Pearson correlation -0.06, p-value 0.74) and showed a weak negative correlation for deletions (Pearson correlation 0.32, p-value 0.05) (*Figure 7—figure supplement 3*). We identified a total of 183 and 62 'unique' duplications and deletions, respectively, when clustering each variant type across all samples based on chromosomal location (see Materials and methods, *Supplementary file 7*). Approximately half the CNVs were located at the chromosome ends, that is 22% and 26% starting within 15 kb of chromosome 5' and 3' ends, respectively. The majority of large CNVs, were present in only a single sample, but some were much more widespread – the most frequent being present in 42 different samples and one variant being present in eight different groups (*Figure 7—figure supplement 4A*). We were particularly interested in CNVs that were present in multiple groups or both species, as these must either have been segregating over a long period of time, or have arisen multiple times independently in different populations. 28% (69 of 245) of all variants were present in both species (*Figure 7—figure supplement 4B*; *Supplementary file 7*) and we investigated those in more detail. We excluded terminal CNVs that showed a gradual coverage increase towards the ends (e.g. *Figure 7—figure supplement 5*) as these have been suspected to be due to telomeric amplifications (*Bussotti et al., 2018*). Several other shared CNVs may represent collapsed repeat regions in the reference genome assembly at which the repeat number varies between samples or where coverage is close to our CNV coverage calling thresholds (e.g. *Figure 7—figure supplement 6*), so we inspected these manually.

We describe in detail two examples of clear CNVs, one deletion and one duplication. The 25 kb long deletion on chromosome 27 was present in 15% of all samples and across four of the different identified groups including both species (*Figure 7A*, deletion 150 in *Supplementary file 7*). It always occurred on a disomic background resulting in the loss of the allele. The 17 genes present in the deleted region were enriched for the GO term 'cilium-dependent motility' due to a single gene annotated as a 'radial spoke protein 3' (LINF_270011200 v41, LinJ.27.2550 v38) (*Figure 7C*). However, other genes including a putative amastin (LINF_270011400 v41, LinJ.27.2550 v38) – part of a large gene family that has an essential role during infection of the mammalian host (*de Paiva et al., 2015*) – were also present in this region. The duplication found on chromosome 35 was only present in a single sample in each, the Ldon1 and Linf1, group and overlapped with the CD1/LD1 locus (*Figure 7B*; duplication 215 in *Supplementary file 7*). In Ldon1, it showed a 2-copy increase on a disomic background, suggesting it was either homozygous for a duplication haplotype or heterozygous with one normal and one 2-copy duplication haplotype. In contrast, the sample from Linf1 has

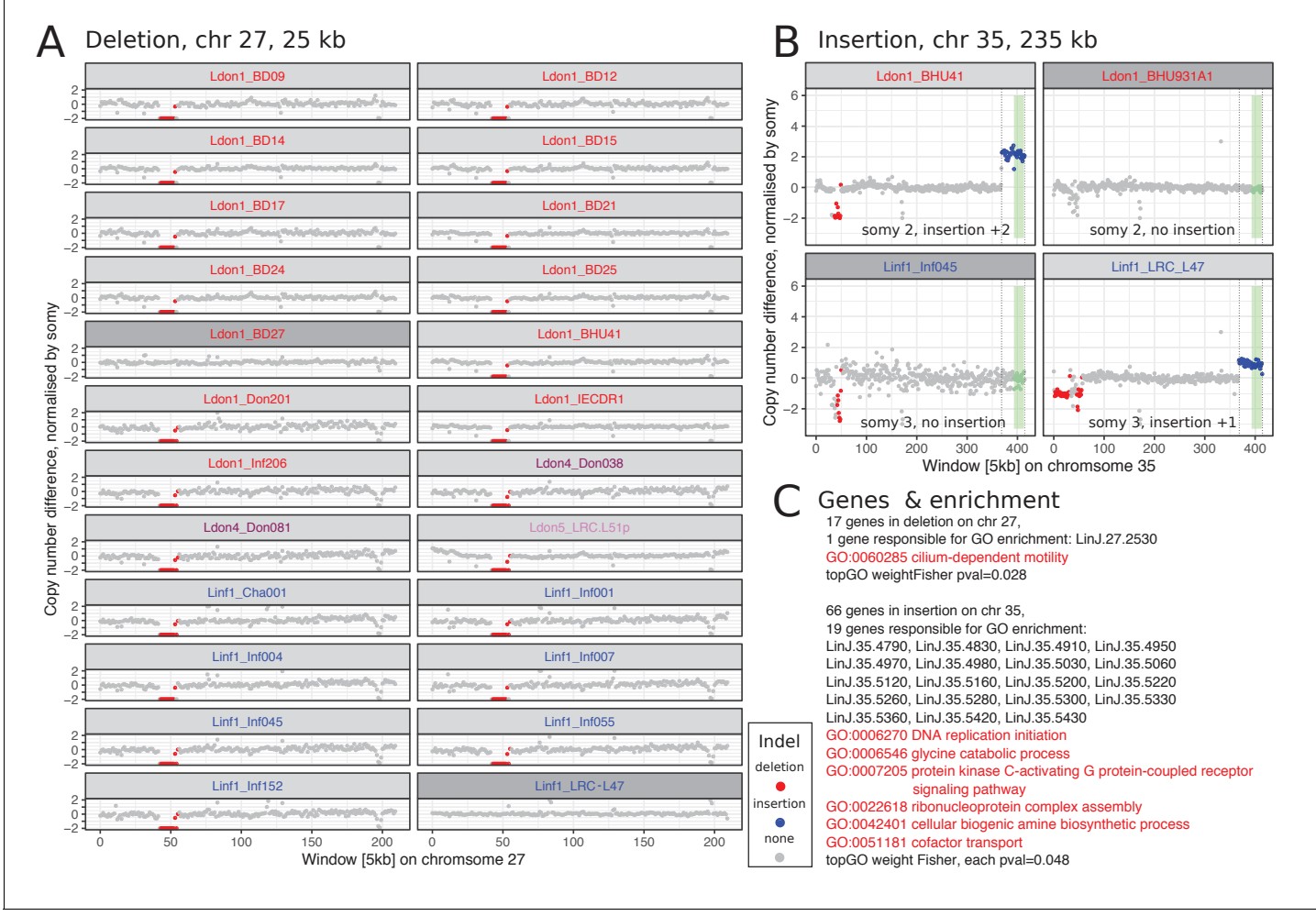

**Figure 7.** Two large CNVs that are shared between both species. (**A**) Chromosome 27 has a 25 kb long deletion that is present in 15% of all samples and four different groups. All chromosomes 27 that have this deletion in our dataset are diploid and the deletion results in a loss of this allele in the respective sample. (**B**) The duplication on chromosome 35 is 235 kb long and present in one isolate of group Ldon1 and Linf1, respectively. The insertion is once present on a disomic background with a 2-fold increase and once on a trisomic background with a 1-fold increase. The green rectangle marks the CD1/LD1 locus sequences for *L. infantum* described in *Sunkin et al. (2001)* (*Supplementary file 8*). For A) and B) a few closely related samples not harbouring the respective CNV are also displayed and highlighted in dark grey. Group identities are indicated by colours of the isolate name. (**C**) Genes present in the respective CNV along with GO enrichment results using topGO (*Alexa et al., 2006*). Details on both CNVs can be found in *Supplementary file 7*: unique CNVs with ids 150 and 215, respectively. The CNV characterisation of the corresponding isolates can be found in *Supplementary file 6*.

The online version of this article includes the following figure supplement(s) for figure 7:

**Figure supplement 1.** Length distribution of large CNVs by chromosome.
**Figure supplement 2.** Most chromosome scale CNVs are located on chromosome 35.
**Figure supplement 3.** Fraction of large CNVs across chromosomes.
**Figure supplement 4.** Large CNVs shared across samples and groups.
**Figure supplement 5.** Increased coverage of samples towards chromosome ends.
**Figure supplement 6.** Indication of a putative assembly error in the reference genome.
**Figure supplement 7.** CNV association with repeat sequences in the genome.
**Figure supplement 8.** Identification of novel repeated sequences on chromosome 27.

a single copy duplication on a trisomic background. 66 genes are present in the insertion enriched for several GO categories (*Figure 7C*). As in *Leishmania* deletions and duplications have been shown to be mediated by repeat sequences (*Ubeda et al., 2014*; for example *Carnielli et al., 2018*), we also looked for previously described and newly identified repeated sequences around the

breakpoint regions of the CNVs on chromosomes 27 and 35 (JPCM5, TriTrypDB v38, RRID:SCR_007043; *Figure 7—figure supplement 7*). For the common deletion on chromosome 27, a few repeats were present close to the 3' and 5' borders of the deleted sequence, respectively. However, no matching repeats were present at both breakpoints that could explain the deletion by the previously described mechanism (*Ubeda et al., 2014*). The large CNVs on chromosome 35 mainly occurred at chromosome ends. We inspected three intra-chromosomal breakpoint regions in a total of five strains, but only in one strain the insertion breakpoint coincided with a repeated sequence (sample LRC_L47, insertion 215, *Figure 7—figure supplement 7*).

To investigate smaller CNVs, we determined the copy number (CN) for each gene in every sample by normalising the median gene coverage by the haploid coverage of the respective chromosome (see Materials and methods). CN variation affected 91.5% of genes (7,625 / 8,330; *Figure 8A*, *Supplementary file 9*), but most CNVs are rare (*Figure 8A*). Only 3.6% of all genes (304) showed a median copy number change ( ≤ -1 or ≥ 1) across samples with 103 genes decreased and 201 increased, respectively (*Figure 8B*). Enrichment tests for the 103 genes with frequently reduced copy number showed GO term enrichments for the biological processes "cation transport", "transmembrane transport", "fatty acid biosynthesis" and "localization" (median CN change across

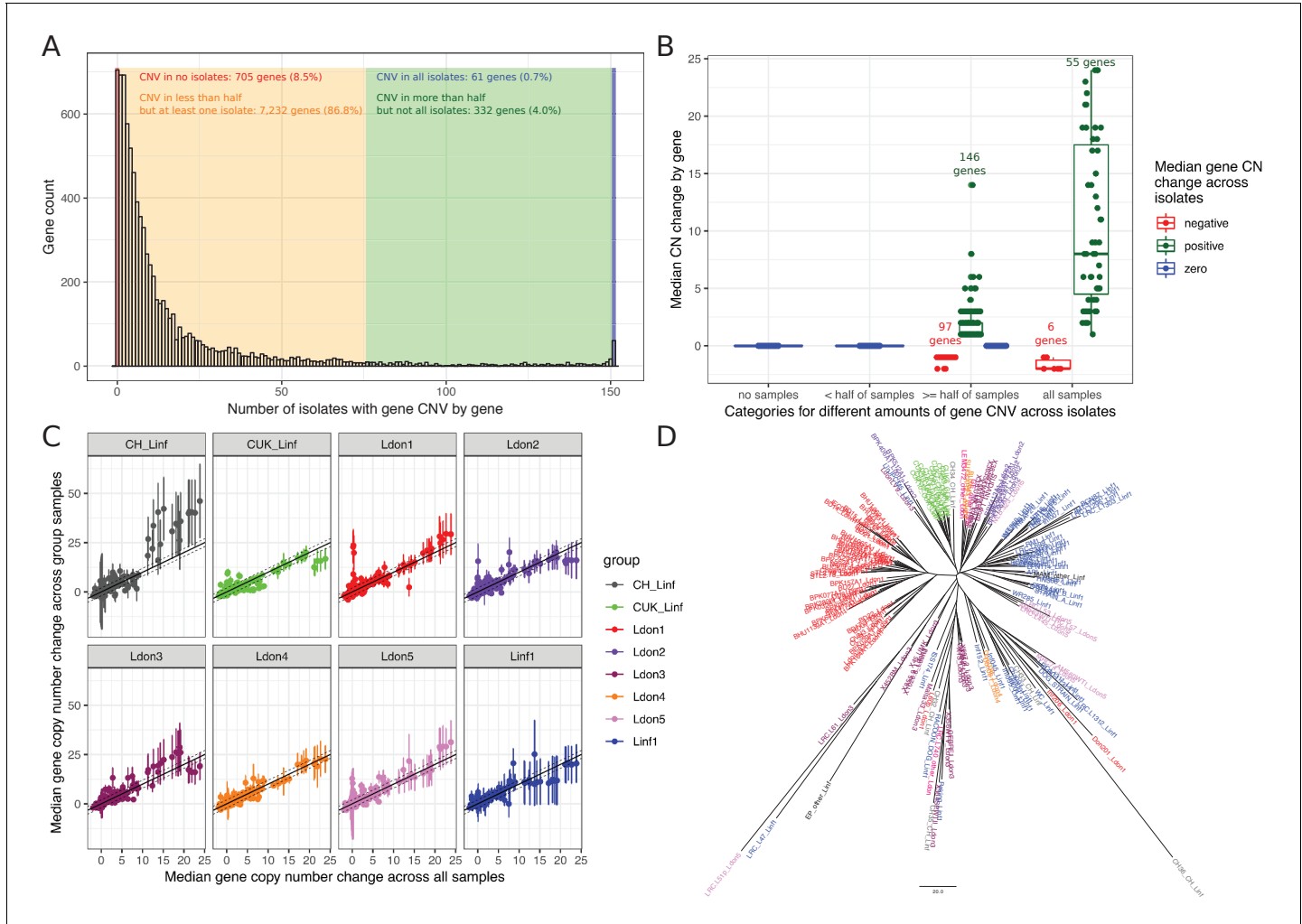

**Figure 8.** Gene copy number variation across groups. (**A**) CN abundances by gene across all 151 isolates. Genes are grouped in four categories (identified by different colours) depending on how many isolates are affected by CN variation in the respective gene. (**B**) Median copy number changes for each gene are shown (individual dots) and summarised for the four different categories also used in sub-figure A including the direction of effect sizes using boxplots. (**C**) Correlations of the median gene copy number across all samples and each respective phylogenetic group. (**D**) Neighbour joining tree using gene CN profiles for each sample.

samples ≤ 1, *Supplementary file 10*). The 201 genes that were regularly increased showed enrichment for several terms including but not exclusive to "modulation by symbiont of host protein kinase-mediated signal transduction", "cell adhesion" and "drug catabolic process" (median CN change across samples ≥ 1, for full list see *Supplementary file 10*). Only a subset of 52 genes (0.6%) showed frequently high gene copy number increases (median ≥ 4 across all samples). Enriched GO terms largely overlapped with enrichments of genes including small CN increases with the additional enrichment of "response to active oxygen species" (*Supplementary file 10*). Those categories might indicate functions on which there is frequent or strong selection pressure. Median gene copy number was positively correlated among groups (*Figure 8C*, Pearson correlation for pairwise comparisons between 0.8 and 0.91). Despite this extensive variation and shared copy number variation across groups, gene copy number still retained some phylogenetic signal (*Figure 8D*).

## Genetic variation for known drug resistance loci

We investigated how genetic variation previously associated with drug resistance is distributed across our global collection of isolates, including loci involved in resistance to or treatment failure of antimonial drugs and Miltefosine (*Table 2*).

The best-known genetic variant associated with drug resistance in *Leishmania* is the so-called H-locus: amplification of this locus is involved in resistance to several unrelated drugs including antimonials (*Callahan and Beverley, 1991*; *Dias et al., 2007*; *Grondin et al., 1993*; *Leprohon et al., 2009*; *Marchini et al., 2003*). In our dataset, the four genes at this locus had an increased gene copy number in 30% of the samples (CN +1 to +44) and a reduced copy number in 9% (CN −1; *Table 2*). 36% of all isolates had a copy number increase of varying degree with identical insertion boundaries that included the genes YIP1, MRPA and argininosuccinate synthase (*Figure 9A*, *Figure 9—figure supplement 1A*, *Table 2*). This duplication was only present in groups Ldon1 and Ldon3 with median increases of approximately +4 and +2, respectively. This matches the rationale that parasites on the Indian subcontinent (largely Ldon1) have experienced the highest drug pressure of antimonials in the past and are suggested to be preadapted to this drug (*Dumetz et al., 2018*) and therefore have the highest prevalence and extent of CN increase, followed by isolates from Sudan and Ethiopia (largely Ldon3). Under this scenario, the Pteridine reductase 1 gene at the H-locus may not be relevant for the drug resistance as it does not show an increased gene CN along with the other genes at that locus (*Figure 9A*). One other isolate, LRC-L51p (Ldon5, India, 1954), had a much larger duplication in this region including the entire H-locus and spanning >45 kb with an enormous increase of ~+44 suggesting an independent insertion or amplification mechanism (*Figure 9—figure supplement 1A*). Four additional isolates showed a copy number increase for only two of the genes at the locus, with different boundaries but always including the MRPA gene (*Figure 9—figure supplement 1B*).

Differential expression of the Mitogen-activated protein kinase 1 (MAPK1) has previously been associated with antimony resistance. However, while (*Singh et al., 2010*) suggested that overexpression is associated with resistance, (*Ashutosh et al., 2012*) suggest the opposite effect potentially implicating an impact of the genetic background. As expression in *Leishmania* is typically tightly linked with gene copy number (*Prieto Barja et al., 2017*; *Iantorno et al., 2017*), we summarised MAPK1 CNVs in our dataset (*Table 2*). 45% of all isolates had an amplified copy number at this locus, including all isolates of Ldon1 and Ldon3 with the highest copy number increase in Ldon1 isolates of between 12 and 41 copies (*Figure 9—figure supplement 2A*, *Table 2*, *Supplementary file 6*). Only a single *L. infantum* isolate had a reduced copy number of one. Increased copy number of MAPK1 is thus associated with isolates from geographical locations with high historical antimonial drug pressures such as the Indian subcontinent and to a lesser extend Africa. Another protein, the membrane channel protein aquaglyceroporin (AQP1), is known to be involved in the uptake of pentavalent antimonials: reduced copy number and expression have been associated with drug resistance (*Andrade et al., 2016*; *Gourbal et al., 2004*; *Monte-Neto et al., 2015*; *Mukherjee et al., 2013*), as has other genetic variation at this locus (*Imamura et al., 2016*; *Monte-Neto et al., 2015*; *Uzcategui et al., 2008*). In our dataset, copy number at this locus was reduced in 6% and increased in 35% of all isolates with small effect sizes (CN −2 to −1 and +1 to +3) but at least one copy of the locus was always present (*Figure 9—figure supplement 2B*, *Table 2*). This may reflect resistance levels in the different populations, while keeping in mind that structural variants generally have a

**Table 2.** Summary of genetic variation across 151 isolates of the *L. donovani* complex for previously described loci involved in resistance or treatment failure of antimonial drugs and Miltefosine.

| locus/ complex | gene id | | | gene name | function prediction | involved in resistance (R)/ treatment failure (TF) to drug: | reference | evidence from reference | gene copy number (gene CN) |
| | *L. infantum, JPCM5, v41* | *L. infantum, JPCM5, v38* | *L. donovani ortholog, BPK282A1, v41* | | | | | | |
|---|---|---|---|---|---|---|---|---|---|
| H-locus | LINF_230007700 | LinJ.23.0280 | LdBPK_230280 | terbinafine resistance gene (HTBF), (YIP1) | | Antimonials (R) | *Callahan and Beverley, 1991*; *Dias et al., 2007* | The *Leishmania* H region is frequently amplified in drug-resistant lines and is associated with metal resistance (genes YIP1, MRPA, PTR1). | Genes have an increased CN in 30% (CN +1 to +44), and reduced CN in 9% (CN −1). 37% of all samples have an insertion including at least three genes (always YIP1, MRPA and argininosuccinate synthase). These amplifications are in groups Ldon1 (42/45), Ldon3 (13/19) and Ldon5 (1/8). The insertion boundaries in isolates from groups Ldon1 and Ldon3 are shared (*Figure 9—figure supplement 1A*). |
| | LINF_230007800 | LinJ.23.0290 | LdBPK_230290 | P-glycoprotein A (MRPA); pentamidine resistance protein 1 | ATP-binding cassette (ABC) transporter, ABC-thiol transporter | Antimonials (R) | *Callahan and Beverley, 1991*; *Dias et al., 2007*; *Leprohon et al., 2009* | Increased expression of MRPA is often due to the amplification of its gene in antimony-resistant strains. | |
| | LINF_230007900 | LinJ.23.0300 | LdBPK_230300 | | argininosuccinate synthase - putative | Antimonials | *Grondin et al., 1993*; *Leprohon et al., 2009* | | |
| | LINF_230008000 | LinJ.23.0310 | LdBPK_230310 | Pteridine reductase 1 (PTR1) | | Antimonials (R) | *Callahan and Beverley, 1991*; *Dias et al., 2007* | see above, evidence only for H-locus in general | |
| | | | | | | Antifolate (R) | *Vickers and Beverley, 2011* | *Leishmania* salvage folate from their hosts. Thereby folates are reduced by a DHFR (dihydrofolate reductase)-TS (thymidylate synthase) and PTR1. PTR1 can act as a metabolic bypass of DHFR inhibition, reducing the effectiveness of existing antifolate drugs. | |
| Mitogen-activated protein kinase, MAPK1 | LINF_360076200 | LinJ.36.6760 | LdBPK_366760 | LMPK, mitogen-activated protein kinase | protein phosphorylation | Antimonials (R) | *Singh et al., 2010*; *Ashutosh et al., 2012* | Conflicting evidence between up- and down-regulation of Mitogen-Activated Protein Kinase one between different studies. | 45% of all isolates showed an increased CN, with all isolates of Ldon1 andLdon3 being affected and smaller fractions in other *L. donovani* groups (*Figure 9—figure supplement 2A*). |
| Aqua-glyceroporin, AQP1 | LINF_310005100 | LinJ.31.0030 | LdBPK_310030 | Aquaglyceroporin 1, AQP1 | drug transmembrane transport | Antimonials (R) | *Gourbal et al., 2004*; *Uzcategui et al., 2008*; *Monte-Neto et al., 2015*; *Andrade et al., 2016*; *Imamura et al., 2016* | A frequently resistant *L. donovani* population has a two base-pair insertion in AQP1 preventing antimonial transport. Increased resistance with decrease in gene CN or expression, while increase leads to higher drug sensitivity. | Gene CN deletions and insertions of small effect sizes (CN −2 to −1 and +1 to +3) are present in 6% and 35% of isolates but never leading to loss of the locus. |

*Table 2 continued on next page*

*Table 2 continued*

| locus/complex | gene id | | | gene name | function prediction | involved in resistance (R)/treatment failure (TF) to drug: | reference | evidence from reference | gene copy number (gene CN) |
|---|---|---|---|---|---|---|---|---|---|
| | *L. infantum*, JPCM5, v41 | *L. infantum*, JPCM5, v38 | *L. donovani* ortholog, BPK282A1, v41 | | | | | | |
| Miltefosine transporter and associated genes | LINF_130020800 | LinJ.13.1590 | LdBPK_131590 | Miltefosine transporter, LdMT | phospholipid transport | Miltefosine (R) | *Pérez-Victoria et al., 2006*; *Shaw et al., 2016* | Gene deletion or different changes in two different strains evolved in promastigote culture for Miltefosine resistance. strain Sb-S: locus deletion and A691P; strain Sb-R: E197D | 15 isolates: +1 gene CNV (CUK, Ldon1, Ldon2, Ldon3, Ldon5) |
| | LINF_130020900 | LinJ.13.1600 | LdBPK_131600 | hypothetical protein | unknown function | Miltefosine (R) | *Shaw et al., 2016* | Deleted along with the Miltefosine transporter gene in a single line evolved for Miltefosine resistance in promastigote culture. | three isolates: +1 gene CNV (Ldon1, Linf1) |
| | LINF_320015500 | LinJ.32.1040 | LdBPK_321040 | Ros3, LdRos3 | Vps23 core domain containing protein - putative | Miltefosine (R) | *Pérez-Victoria et al., 2006* | Putative subunit of LdMT; LdMT and LdRos3 seem to form part of the same translocation machinery that determines flippase activity and Miltefosine sensitivity in *Leishmania*. | one isolate: +1 gene CNV (Ldon1) |
| Miltefosine sensitivity locus, MSL | LINF_310031200 | LinJ.31.2370 | LdBPK_312380 | | 3'-nucleotidase/nuclease - putative | Miltefosine (TF) | *Carnielli et al., 2018* | MSL: a deletion of this locus was associated with Miltefosine treatment failure in Brazil. While the frequency of the MSL was still relatively high in the North-East it was almost absent in the South-East of Brazil, and it was absent in *L.infantum/L. donovani* in the Old World. | Genes have a reduced CN in 55% (CN −1 to −8) and increased in 4% (CN +1). Four isolates, show a complete loss of the MSL at identical boundaries: WC, Cha001, HN167 and HN336 (2/4 isolates from Brazil, 2/2 isolates from Honduras). Two isolates show a reduction of all four genes at this locus but with various deletion boundaries: IMT373cl1 (Portugal), CH35 (Cyprus) (*Figure 9B*). |
| | LINF_310031300 | LinJ.31.2380 | LdBPK_312380 | | 3'-nucleotidase/nuclease - putative | Miltefosine (TF) | *Carnielli et al., 2018* | | |
| | LINF_310031400 | LinJ.31.2390 | LdBPK_312390 | | helicase-like protein | Miltefosine (TF) | *Carnielli et al., 2018* | | |
| | LINF_310031500 | LinJ.31.2400 | LdBPK_312320, LdBPK_312400 | | 3–2-trans-enoyl-CoA isomerase - mitochondrial precursor - putative | Miltefosine (TF) | *Carnielli et al., 2018* | | |

chance to get lost during in vitro culturing as experienced by our samples (e.g. see *Domagalska et al., 2019*).

The Miltefosine transporter in *L. donovani* (LdMT) together with its putative ß subunit LdRos3 have been shown to be essential for phospholipid translocation activity and thereby the potency of the anti-leishmanial drug Miltefosine (*Pérez-Victoria et al., 2006*). In a drug selection experiment, Miltefosine resistant parasites showed common and strain-specific genetic changes including deletions at LdMT and single base mutations (*Shaw et al., 2016*). Neither LdMT, Ros3 or a hypothetical protein deleted together with LdMT in a drug selection experiment (*Shaw et al., 2016*), showed a reduction in gene copy number across our 151 isolates (*Figure 9—figure supplement 2C*, *Supplementary file 9*). Moreover, no SNP variation was present in two codons (A691, E197; *Shaw et al., 2016*) putatively associated with drug resistance (*Table 2*). The Miltefosine sensitivity locus (MSL) was recently identified as a deletion associated with treatment failure in a clinical study of patients with VL in Brazil (*Carnielli et al., 2018*). In the same study, further genotyping of the MSL showed clinal variation in the presence of the locus ranging from 95% in North East Brazil to <5% in the South East (N = 157), while no deletion was found in the Old World. The entire locus including all four genes (*Table 2*) was completely deleted in four of our samples of the Linf1 group including two of the four samples from Brazil (Cha001 1974, WC 2007) and in the two samples from Honduras (HN167 1998, HN336 1993) (*Figure 9B*, *Supplementary file 9*) with deletion boundaries coinciding with those reported previously (*Carnielli et al., 2018*). Another isolate, IMT373cl1 (collected in Portugal, 2005) showed a deletion of a larger region (90 kb), reducing the local chromosome copy

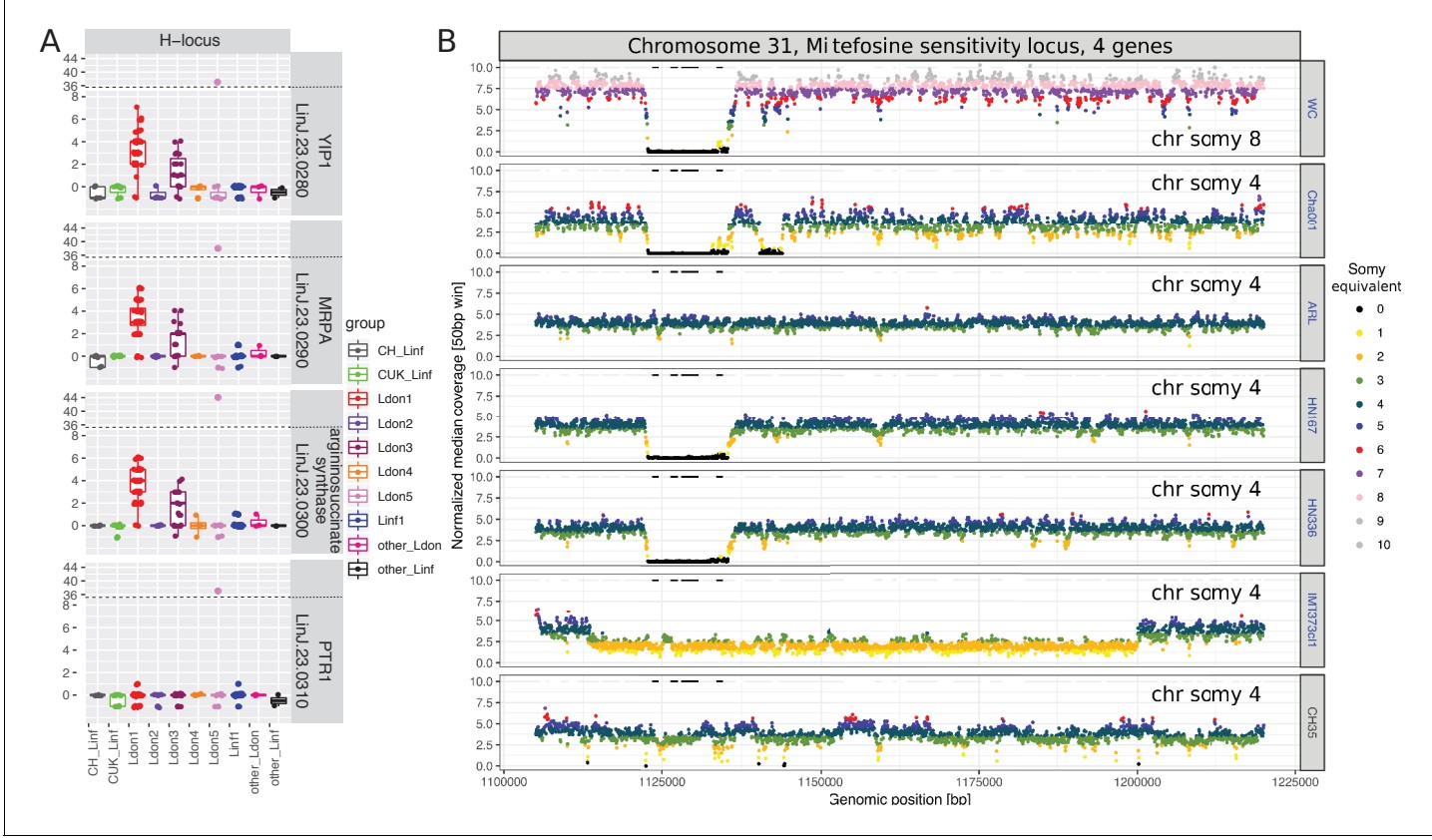

**Figure 9.** Copy number variation of putative drug resistance genes. (A) Copy numbers (CNs) for all four genes on the H-locus are shown for all 151 samples across all 10 different (sub-)groups. (B) Genome coverage in the genomic regions surrounding the MSL in all six samples showing a deletion and one sample with no CN reduction. Genome coverage for 50 bp windows is normalised by the haploid chromosome coverage and colours indicate the somy equivalent coverage of the respective window. The genes, LinJ.31.2370, LinJ.31.2380, LinJ.31.2390 and LinJ.31.2400, are marked as black horizontal lines. Colours of the sample names indicate group colours used throughout this study.

The online version of this article includes the following figure supplement(s) for figure 9:

**Figure supplement 1.** Copy number increase at the H-locus.
**Figure supplement 2.** Copy number variation of putative drug resistance genes.
**Figure supplement 3.** Measures of adaptive evolution.
**Figure supplement 4.** Gene ontology enrichment of marker genes with putative biological impact.

number from four to two (*Figure 9B*). The sixth sample that showed a copy number decrease of all four MSL associated genes, only showed a marginal and variable reduction in coverage and might be better explained by noise in genome coverage (*Figure 9B*).

## Population and species-specific selection

We investigated putative species-specific selection, summarizing selection across the genome using the numbers of fixed vs. polymorphic and synonymous vs. non-synonymous sites for each species across all genes: The α statistic, originally by *Smith and Eyre-Walker (2002)*, is a summary statistic, presenting the proportion of non-synonymous substitutions fixed by positive selection and is often used to summarize patterns of selection in a species. In both, *L. donovani* and *L. infantum*, α was negative, with −0.19 and −0.34, respectively, showing an excess of non-synonymous polymorphisms but lacking a clear biological interpretation. Out of 8234 genes tested for departure of neutrality using the McDonald-Kreitman test, only two and four genes showed signs of positive selection (p-value<0.05, FDR = 1) and 11 and 12 an excess of non-synonymous differences (p-value<0.05, FDR = 1) for *L. donovani* and *L. infantum*, respectively (*Figure 9—figure supplement 3*, *Supplementary file 11*). Interestingly, one of the genes with putative signs of adaptive evolution in *L. donovani* (LINF_330040400 v41, LinJ.33.3220 v38) was previously associated with in vivo enhanced

virulence and increased parasite burden in vitro for *L. major* when overexpressed (*Reiling et al., 2010*). In our dataset, this gene contained nine missense, 3 synonymous and 19 upstream/intergenic SNP-variants differentially fixed between *L. donovani* and *L. infantum* (*Supplementary file 4*), which might provide further candidates for differences in virulence between both species.

While genetic variants can become fixed in different populations by either neutral forces (genetic drift) or positive selection, we took advantage of the genetic differentiation between groups to search for group-specific SNPs that might be of biological relevance. We investigated whether particular functional categories (biological processes in Gene Ontology) were enriched among genes containing high or moderate effect group- and species-specific SNP variants (*Supplementary file 12*). While most enrichment terms were specific to one marker set, the terms 'protein phosphorylation', 'microtubule-based movement' and 'movement of cell or subcellular component' were enriched in five, three and two out of the nine tested SNP sets, respectively (*Figure 9—figure supplement 4*). More group specific enrichments with potentially more easily interpretable biological implications include 1) 'response to immune response of other organism involved in symbiotic interaction' for Ldon1, 2) 'mismatch repair' for Linf1 in response to oxidative stress and 3) 'pathogenesis' for the *L. infantum – L. donovani* species comparison (*Figure 9—figure supplement 4*). For the species comparison, the enrichment of the term 'pathogenesis' was due to fixed differences of putative functional relevance in genes including a protein containing a Tir chaperone (CesT) domain, a subtilisin protease and a Bardet-biedl syndrome one protein that are putative candidates for increased pathogenicity in *L. donovani* (*Table 3*, *Supplementary file 4*). Tir (translocated intimin receptor) chaperones are a family of key indicators of pathogenic potential in gram-negative bacteria, where they support the type III secretion system (*Delahay et al., 2002*). Proteins containing these domains are almost exclusive to kinetoplastids among eukaryotes. In *L. donovani*, a subtilisin protease (SUB; Clan SB, family S8), has been found to alter regulation of the trypanothione reductase system, which is required for reactive oxygen detoxification in amastigotes and to be necessary for full virulence (*Swenerton et al., 2010*). The Bardet-biedl syndrome 1 (BBS1) gene in *Leishmania* was shown to be

**Table 3.** Candidate genes putatively involved in pathogenesis associated differences between *L. donovani* and *L. infantum*. Candidates were identified through GO enrichment analysis of moderate to high effect variants between both species across our 151 isolates.

| Gene name | Gene codes v41 (v38) TritrypDB (http://tritrypdb.org/tritrypdb/) | Annotation | Fixed genomic variation between *L. infantum* and *L. donovani* (changes *L.inf* > *.don*) | Evidence for pathogenic function |
|---|---|---|---|---|
| Tir chaperone protein | LINF_040012200 (LinJ.04.0710), LINF_340038600 (LinJ.34.2950) | Tir chaperone protein (CesT) family/PDZ domain containing protein - putative, Tir chaperone protein (CesT) family - putative | nt 362A > G; aa Glu121Gly<br>nt 594A > G; aa Gln198Gln<br>nt 1659A > C; aa Lys553Asn<br>nt 1703A > G; aa Asn568Ser | Part of secretion system to deliver virulence effector proteins into the host cell cytosol in gram-negative bacteria; secreted proteins require chaperones to maintain function (*Delahay et al., 2002*). |
| Subtilisin protease | LINF_130015300 (LinJ.13.0940 and LinJ.13.0930*[1], -strand, are fused in v41 with an extra 54 bp in between them) | subtilisin-like serine peptidase | nt 2813T > G; aa Phe938Cys<br>nt 3346G > A; aa Gly1116Ser<br>nt 4389G > A; aa Pro1463Pro*<br>nt 5014A > C; aa Ser1672Arg* | Shown to be essential for full virulence and involved in detoxification of ROS in *L. donovani* (*Swenerton et al., 2010*). |
| Bardet-biedl syndrome one protein | LINF_350047600 (LinJ.35.4250) | Bardet-Biedl syndrome one protein homolog (BBS1-like protein 1) - putative | nt 531C > T; aa Ser177Ser<br>nt 580G > A; aa Ala194Thr<br>nt 1038C > A; aa Arg346Arg<br>nt 1221T > C; aa Gly407Gly<br>nt 1310C > T; aa Ala437Val | *Leishmania* BBS1 knock-out mutants have reduced infectivity for in vivo macrophages and infection of BALB/c mice was severely compromised (*Price et al., 2013*). |

*Nucleotide (nt) and amino acid (aa) changes in LinJ.13.0930*[1] (v38) have been adapted to positions to its fused version LINF_130015300 (v41) in this table. Positions for v38 can be found in **Supplementary file 4**.

involved in pathogen infectivity. BBS1 knock-out strains, as promastigotes in vitro, had no apparent defects affecting growth, flagellum assembly, motility or differentiation but showed a reduced infectivity for in vitro macrophages and the ability to infect BALB/c mouse of null parasites was severely compromised (*Price et al., 2013*).

## Discussion

Our whole-genome sequence data represents much of the global distribution of the *L. donovani* species complex. Compared to previous genomic studies on the *L. donovani* complex that focused on more geographically confined populations (*Carnielli et al., 2018*; *Downing et al., 2011*; *Imamura et al., 2016*; *Rogers et al., 2014*; *Teixeira et al., 2017*; *Zackay et al., 2018*), our sampling revealed a much greater genetic diversity. We identified five major clades of *L. donovani* that largely reflect the geographical distribution of the parasites and their associated vector species (*Akhoundi et al., 2016*). Some, such as the Middle Eastern group (Ldon4) are within themselves diverse, and in this case represented by a few samples, suggesting that a deeper sampling of parasites in this region may be needed. In contrast, our data confirmed that the low diversity of the main genotype group from the Indian subcontinent (*Imamura et al., 2016*) is indeed unusual, which might be related to the epidemic nature of VL on the Indian subcontinent (*Dye and Wolpert, 1988*). The main *L. infantum* clade is widespread and displays little diversity, although two subgroups represent the classical MON-1 and non-MON-1 Mediterranean lineages which co-segregate in the same geographical regions interfering with isolation-by-distance relationships in that group (*Figure 1A*, *Figure 1—figure supplement 1*). Our data highlighted some weaknesses in previous typing systems for characterising *Leishmania* using MLEE (*Rioux et al., 1990*) and MLMT (*Schönian et al., 2011*; *Schönian et al., 2008*). We confirmed paraphyly of the zymodeme MON-37 across *L. donovani* groups (see also *Alam et al., 2009*) and for the zymodemes MON-30 and MON-82 within the Ldon3 group (*Figure 1—figure supplement 1*). Moreover, the MON-1 zymodeme groups together parasites from the Mediterranean region and South America but also a sample from the genetically distinct Asian subgroup (*Figure 1—figure supplement 1*). While data from MLMT (e.g. *Kuhls et al., 2007* and *Gouzelou et al., 2012*) is much more congruent with our results, we explain diversity within the previously assigned Cypriot population (*Gouzelou et al., 2012*) by hybridisation of some of these isolates (*Figures 1A* and *4*, *Figure 4—figure supplement 1A*) and also describe hybridisation in other groups (e.g. LEM3472, GE and LRC-L740) that was not apparent with microsatellite markers (*Kuhls et al., 2007*).

Two regions emerged as apparent hot-spots of diversity in this species complex. The first is the Eastern Mediterranean, where the high genetic diversity of parasites assigned to *L. infantum* appears to be driven by hybridisation between *L. infantum* from China and a genotype identified in Cyprus (i.e. CH33, 35 and 36) (*Figure 4—figure supplement 8*). This gave rise to the isolates from Çukurova described previously (*Rogers et al., 2014*) and some other hybrid genotypes from Cyprus (CH32 and 34) and suggests parasite movement from Central Asia/China to the Eastern part of the Mediterranean in the relatively recent past. The phylogenetic origin of the five Cypriot isolates has been unclear: they were placed in the paraphyletic zymodeme MON-37 of *L. donovani* (*Antoniou et al., 2008*) but clustering based on microsatellite profiles placed them in a clade of *L. infantum* between zymodeme MON-1 and non-MON-1 isolates (*Gouzelou et al., 2012*). Our data supports a deep-branching clade of CH and CUK isolates distinct from other isolates of *L. infantum* (*Figure 1A*, *Figure 1—figure supplement 1*) but the precise phylogenetic position of this group varies somewhat for different parts of the genome (*Figure 4—figure supplement 8B*). The origin of the pure, that is 'non-hybrid' Cypriot samples (CH33, 35, 36), however, is not completely resolved: they could be either a distinct evolutionary linage within the *L. donovani* complex, or ancient hybrids between *L. infantum* and *L. donovani*. The other geographical regions of high diversity within the *L. donovani* complex is further South, encompassing the horn of Africa, the Arabian Peninsula and adjacent areas of the Middle East. Some of this diversity has been reported showing the presence of two clearly distinct groups of *L. donovani*: one in North-East and the other one in East Africa (*Zackay et al., 2018*). This genetic differentiation between both populations corresponds to their geographic separation by the rift valley in Ethiopia with different ecology and vector species (*Gebre-Michael et al., 2010*; *Gebre-Michael and Lane, 1996*) but hybrids between these populations have also been described (*Cotton et al., 2019*). More striking is the high diversity of *L. donovani* lineages in the Arabian

Peninsula and the Middle East, including lineages present on both sides of the Red Sea and hybrids between groups present in this region and Africa (Ldon4 and other Ldon). The Middle East and adjacent regions may represent a contact zone where European, African and Asian lineages meet and occasionally hybridise increasing local genetic diversity. Moreover, the hybrid samples GE, LEM3472 and LRC-L740 sampled in Sudan and Israel with putative parental ancestry from Sudan/Ethiopia (Ldon3) as well as the Middle East (Ldon4) also suggest relatively recent parasite movements between those geographical regions. More extensive sampling in both of these 'hot-spot' regions would likely further improve our knowledge of the genetic diversity and geographic movements within the *L. donovani* species complex. Besides these 'diversity hot-spots', many other regions were sparsely sampled for our data collection and are under-explored by *Leishmania* researchers in general. While we have few isolates in our main analysis from the New World, where VL is present in much of Central America, and northern South America, we show that a total of 31 'MON-1' samples from Central/South America are closely related and likely of South-Western European origin. Two different lineages (i.e. MON-1 and non-MON-1) containing European as well as American *L. infantum* also suggest at least two introductions of the parasite into the New World (*Figure 1—figure supplement 2A*), which are also broadly consistent with suggested ancient changes in the geographical distribution of the species complex (*Lukes et al., 2007*). Our sampling, however, remains sparse in Central Asia, where both *L. infantum* and *L. donovani* may be present. From China we only have *L. infantum* isolates, but there is likely to be a diverse range of *L. donovani*-complex parasites present (*Alam et al., 2014*; *Zhang et al., 2013*).

While we identified many novel lineages that are hybrids between major groups present in our study, it is likely that even with whole-genome variation data we are missing other admixture events especially within groups: This is because admixture analysis is most suited to identify admixed samples between the given *K* groups, and heterozygosities are most prominent when hybridisation occurs between genetically diverse strains. All of our known hybrid populations had elevated levels of heterozygosity, but group Ldon3 was highly heterozygous without distinct genomic patterns of hybridisation (*Figure 3A*). Clear genomic patterns of hybridisation can be undetectable when hybridisation occurs frequently between closely related strains. This might be the case for the Ldon3 group and is also supported by a steep decline of LD with genomic distance (*Figure 5*) and the mixed distribution of isolate specific haplotypes within the Ldon3 group (*Figure 4—figure supplement 9B–D*). However, while we don't have direct proof of hybridisation in the Ldon3 group, the generality of the relationship between heterozygosity and hybrid origin remains unclear. We investigated evidence for hybridisation from the admixture analysis (*Figure 1A*) at a range of values of the parameter *K* (the number of distinct populations present in the data; *Figure 4—figure supplement 6*), also considering that many of the assumptions of admixture analysis are likely not to hold in *Leishmania* populations. However, this approach missed the known hybrids of the Çukurova population, which were consistently identified as a separate, 'pure' population (*Figure 4—figure supplement 6*). Therefore, we used an approach similar to that used by *Rogers et al. (2014)* to identify genome regions that seem to be homozygous for each of the two putative parental groups of the hybrids. While this haplotype-based approach could identify parents of the Çukurova isolates, it did not clearly resolve the origins of other samples suggested to be hybrid by the admixture analysis. This could be either because our sample collection does not include the parental lineage or a close relative, or because these samples are of much older hybrid origin, so that subsequent recombination has erased the haplotype block structure we are looking for (e.g. see *Rogers et al., 2014*). Different approaches are therefore needed to investigate recombination within populations. We also used the level of linkage disequilibrium and particularly the decrease in LD with distance as an indicator of recombination to show that the impact of recombination differs greatly between *L. donovani* complex populations. However, LD is a complex measure affected by a range of other factors including population structure and demographic factors (*Slatkin, 1994*), so we cannot directly quantify recombination rates from observed patterns of LD in *Leishmania*. Additionally, we observed major differences in the allele frequency spectrum in different populations, in agreement with putative recombination differences and the unique evolutionary history of each group.

The variation in coverage between chromosomes and unusual allele frequency distributions in our isolates (*Figure 2—figure supplement 2*) confirmed the presence of extensive aneuploidy in our samples, as observed for all *Leishmania* promastigote cultures investigated to date. In our study, this variation in aneuploidy between samples reflected differences in the average chromosome copy

number of a population of promastigote cells grown in vitro for each isolate, and showed no apparent phylogenetic structure. We assume that this reflects the well-documented mosaic aneuploidy present across *Leishmania* populations (*Prieto Barja et al., 2017*; *Lachaud et al., 2014*; *Sterkers et al., 2011*), where aneuploidy variation is present between cells within a parasite population. This variation could be selected upon and quickly change mean observed aneuploidies in a new environment, such as in vitro culture. However, we cannot directly address aneuploidy mosaicism with our data due to pooling cells within a strain for sequencing. To address this issue in future studies and understand the dynamics of *Leishmania* aneuploidy in infections and in culture, single-cell approaches seem to be most promising (e.g. *Dujardin et al., 2014*).

Similarly, our data reflects the genetic variability of a set of isolates grown as promastigotes in axenic culture in vitro, a very different environment, and different life stage of the parasite to that present in patients. This means that we may miss variation present within host parasite populations that are lost during parasite isolation or subsequent growth, and that our results may be affected by selection to in vitro environments: In particular aneuploidy patterns in vectors and mammalian hosts were shown to differ from that in culture (*Domagalska et al., 2019*; *Dumetz et al., 2017*), and have other variants in particular during long term in vitro adaptation (e.g. *Sinha et al., 2018*; *Bussotti et al., 2018*). Given the breadth of global isolate collection used in our study it was not possible for us to ensure that common culture conditions were used for all the isolates. A recent approach to directly sequence *Leishmania* genomes in clinical samples has given some first insights into the effects of parasite culture in vitro and will allow future studies of *Leishmania* genome variation to avoid this potential bias (*Domagalska et al., 2019*).

Changes in gene dosage – of which aneuploidy is just the most striking example – have been shown to have a profound impact on gene expression in *Leishmania*, which lacks control of transcription initiation (*Campbell et al., 2003*). We identified extensive copy number variation, including both very large structural duplications and deletions and smaller-scale variants affecting single genes. Large structural variants are particularly common on chromosome 35. Here, eight strains showed a range of large CNVs (30–675 kb; *Figure 7—figure supplement 2*, *Supplementary file 7*) at the 3' end of the chromosome that overlapped with the CD1/LD1 locus previously described as being maintained as extrachromosomal linear or circular molecules of various lengths in several *Leishmania* species (*Lemley et al., 1999*; *Segovia and Ortiz, 1997*; *Tripp et al., 1992*; *Tripp et al., 1991*). Our analysis indicated at least 9 duplications of various lengths containing the CD1/LD1 locus, but our short-read sequencing data was insufficient to reveal the structure/insertion type in the genome. The CD1/LD1 locus is also known to arise spontaneously in independent in-vitro cell lines (*Segovia and Ortiz, 1997*) and encodes the biopterin transporter (*Kündig et al., 1999*). However, whether the CNVs we observed were amplified before or during culturing of our isolates or might provide a growth advantage in certain media would require direct experimental investigation. Many CNVs appeared too widespread across different clades to have evolved neutrally. Particularly a common deletion on chromosome 27 (*Figure 7A*) shared identical breakpoints across 22 samples. As no repeat structures were present at the breakpoints that could explain independent deletion events causing identical breakpoints (*Figure 7—figure supplement 7*), this suggests that the deletion might be an ancient segregating polymorphism. While it is difficult to identify the specific functional relevance of these variants without phenotypic or functional information, these might be interesting targets for future functional studies. Additionally, we demonstrated the utility of genome data to understand functional genetic variation for variants with previously known impacts on phenotypes such as drug resistance. The deletion at the MSL locus, previously associated with Miltefosine treatment failure, is restricted to the New World and was considered to have evolved within Brazil (see also *Carnielli et al., 2018*) but for the first time we reported this variant in Honduras, suggesting a wider geographical wider distribution than previously appreciated. Moreover, varying local frequencies and copy numbers of the H-locus and the MAPK1 duplication in India and North East Africa suggest that resistance against antimonials is more widespread on the Indian subcontinent, and may mediate a higher level of resistance than in other locations.

Our study provides the first comprehensive view of the globally distributed, whole-genome genetic diversity of the two most pathogenic species of *Leishmania* and any *Leishmania* species to date. Our ability to capture a much more comprehensive picture of the genetic variation in these species allowed us to identify differences between species with respect to diversity and isolation-by-distance, reveal the impact of aneuploidy turnover on genetic diversity and showed different

amounts of recombination in different geographical regions. The investigation of CNVs with respect to the role of repeated sequences was shown in a broader genomic context and we identified particular regions as apparent hotspots for the generation of genetic diversity in this species. Moreover, the availability of this broad and deep genomic resource for *L. dononvani* and *L. infantum* has allowed us to identify and understand the ancestry of hybrid strains in many foci. This work provides a valuable resource in investigating individual loci to understand functional variation as well as placing more focused studies into a global context.

## Materials and methods

### Choice of samples and sample origin

The genetic diversity of 151, mostly clinical isolates, from the *L. donovani* complex, and spanning the entire global distribution of this species complex was investigated to reveal the complex's whole-genome diversity on a global scale. This includes 97 isolates that we sequenced specifically for this study, complemented with whole-genome sequence data of 33 isolates from the Indian subcontinent (*Imamura et al., 2016*), 11 from a known Turkish hybrid population (*Rogers et al., 2014*), seven from Ethiopia (N = 1, *Rogers et al., 2011*); N = 6, *Zackay et al., 2018*), two from Sri Lanka (*Zhang et al., 2014*) and the whole-genome sequences of the JPCM5 reference strain (*Peacock et al., 2007*). The samples taken from other studies present a large proportion of all available sequences for *Leishmania* to date. Of regions where the genetic diversity had previously already been described for many samples, we chose subsets representing the known genetic diversity (i.e. *Imamura et al., 2016*; *Zackay et al., 2018*). In an additional analysis (*Figure 1—figure supplement 2B*), we included 26 isolates from three different states in Brazil (*Carnielli et al., 2018*) to confirm reduced genetic diversity in South America. The 97 samples sequenced for this study are deposited in ENA under the study accession numbers: PRJEB2600 (ERP000767), PRJEB2724 (ERP000966), PRJEB8947 (ERP009989) and PRJEB2115 (ERP000169) (https://www.ebi.ac.uk/ena/data/view). All metadata on the 151 isolates including ENA accession numbers of individual samples are summarized in *Supplementary file 1* (see also https://microreact.org/project/_FWlYSTGf; *Argimón et al., 2016*). The promastigote cultures and DNA samples came from different *Leishmania* strain collections: The London School of Hygiene and Tropical Medicine; The Hebrew University, Jerusalem WHO Reference Centre for the Leishmaniases; The Academic Medical Centre (University of Amsterdam), Medical Microbiology, Section Parasitology; The Bangladesh Agricultural University, Mymensingh; The Centre National de Référence des Leishmanioses Montpellier; The Istituto Superiore di Sanità Roma; The Hellenic Pasteur Institute Athens; The Koret School of Veterinary Medicine, Hebrew University, Jerusalem, Israel; The Coleção de *Leishmania* do Instituto Oswaldo Cruz, Rio de Janeiro; The University of Khartoum; The Universitat Autònoma de Barcelona; The Institute of Tropical Medicine Antwerp, and The Charité University Medicine Berlin. Only previously collected isolates from humans and animals have been used in this study. The parasites from human cases had been isolated as part of normal diagnosis and treatment with no unnecessary invasive procedures and data on human isolates were encoded to maintain anonymity.

### Whole-genome sequencing of clinical isolates

The 97 isolates new to this study were grown as in vitro promastigote culture to generate material for sequencing as had been done for the 54 remaining sequenced isolates taken from other sources (*Imamura et al., 2016*; *Peacock et al., 2007*; *Rogers et al., 2014*; *Rogers et al., 2011*; *Zackay et al., 2018*; *Zhang et al., 2014*). Of all these, most (62%) were not cloned and regrown from a single cell before sequencing; 6% of the isolates had been cloned and 32% were of unknown status prior to sequencing (*Supplementary file 1*). Genomic DNA was extracted by the phenol-chloroform method and quantified on a Qubit (Qubit Fluorometric Quantitation, Invitrogen, Life Technologies). DNA was then sheared into 400–600-base pair fragments by focused ultrasonication (Covaris Adaptive Focused Acoustics technology, AFA Inc, Woburn, USA). Standard indexed Illumina libraries were prepared using the NEBNext DNA Library Prep kit (New England BioLabs), followed by amplification using KAPA HiFI DNA polymerase (KAPA Biosystems). 100 bp paired-end reads were generated on the Illumina HiSeq 2000 according to the manufacturer's standard sequencing protocol (*Bronner et al., 2014*).

## Read mapping pipeline

Reads were mapped with SMALT (RRID:SCR_005498, v0.7.4, *Ponstingl, 2010*) using the parameters: '–x –y 0.9 –r 1 –i 1500' specifying independence of paired-end reads, a minimum fraction of 0.9 of matching bases, reporting of a random best alignment if multiple are present and a maximum insert size of 1500 bp against the reference genome JPCM5 of *L. infantum* (MCAN/ES/98/LLM-877, Tri-TrypDB v38, RRID:SCR_007043; *Aslett et al., 2010*). Mapped reads were sorted and duplicate reads were marked with picard 'MarkDuplicates' (RRID:SCR_006525, v1.92, https://broadinstitute.github.io/picard/). For resulting individual bam files per isolate, indels were called and local realignment was performed with GATK using the 'RealignerTargetCreator' and 'IndelRealigner' with default settings (RRID:SCR_001876, v2.6–4, *DePristo et al., 2011*).

## Reference genome masking

We developed a custom mask for low complexity regions and gaps in the reference genome. To identify low complexity regions, we used the mappability tool from the GEM library (release3, *Derrien et al., 2012*). Gem-mappability was run with the parameters -l 100 m 5 -e 0 `--max-big-indel-length` 0 `--min-matched-bases` 100, specifying a kmer length of 100 bp with up to 5 bp mismatches. This gives the number of distinct kmers in the genome, and we calculated the uniqueness of each bp position as the average number of kmers mapping a bp position. Any base with a GEM uniqueness score >1 was masked in the reference genome including a flanking region of 100 bp at either side. This approach masked 12.2% of the 31.9 Mb genome.

## Determination of sample ploidies

To determine individual chromosome ploidies per isolate the GATK tool 'DepthOfCoverage' (RRID: SCR_001876, v2.6–4) was used to obtain per-base read depth applying parameters: '`--omitInter-valStatistics --omitLocusTable --includeRefNSites --includeDeletions --printBaseCounts`'. Results files were masked using our custom mask (see 'Reference Genome Mask'). Summary statistics were calculated per chromosome, including median read depth. The median read depth for each chromosome was used to estimate chromosome copy number, somy, for each sample using an Expectation-Maximization approach previously described in *Iantorno et al. (2017)*. For a few isolates where the coverage model appeared to be overfitting (high deviance values), somy estimates were manually curated by examining both coverage and allele frequency data. Where allele frequency distributions did not support high somy values, they were altered so that the majority of chromosomes were disomic and individual errors were corrected to fit clear somy expectations suggested by the respective allele frequency spectra.

## Somy evaluation based on allele frequency profiles

For isolates with high genome-wide heterozygosity ( > = 0.004) peaks of allele frequency distributions were estimated for chromosomes with at least 100 SNPs using the density function (stats package, *R Development Core Team, 2013*). After peak estimation of allele frequency distributions by isolate and chromosome unreasonable peaks were removed, that is the ones that are too low (smaller than 0.2 of the highest peak). The estimated peak vector for each chromosome and isolate were then compared to peak distributions expected for the respective somy, for example for a diploid, triploid and tetraploid chromosome we expect peaks only at the frequencies $\frac{1}{2}$, $\frac{1}{3}$ & $\frac{2}{3}$ and $\frac{1}{4}$ & $\frac{2}{4}$ & $\frac{3}{4}$, respectively. Deviations were calculated as the sum of square roots of absolute differences to the closest matched peaks of expected peak distributions. Peak estimates are shown in *Figure 4—figure supplement 2* and deviations between coverage and frequency based somy estimates in *Figure 4—figure supplement 5*.

## Variant calling

Variant calling was done following the Genome Analysis ToolKit (GATK, RRID:SCR_001876) best-practice guidelines (*Van der Auwera et al., 2013*) with modifications detailed below. Given the aneuploidy of *Leishmania*, we considered individual somies per chromosome and isolate: the GATK 'HaplotypeCaller' (RRID:SCR_001876, v3.4–0, *DePristo et al., 2011*) was used with the parameters '–sample_ploidy SOMY -dt NONE –annotateNDA' and additionally all-sites files were generated by adding the additional flag '-ERC BP_RESOLUTION' to the above HaplotypeCaller command.

Individual vcf files (by chromosome and isolate) were processed, filtered and combined with custom made scripts implementing the following steps: only SNPs outside masked regions (see 'Reference Genome Masking') were extracted; SNPs were hard filtered excluding genotypes failing to pass at least one of the following criteria: DP >= 5*SOMY, DP <= 1.75*(chromosome median read depth), FS <= 13.0 or missing, SOR <= 3.0 or missing, ReadPosRankSum <= 3.1 AND ReadPosRankSum >= −3.1, BaseQRankSum <= 3.1 AND BaseQRankSum >= −3.1, MQRank-Sum <= 3.1 AND MQRankSum >= −3.1, ClippingRankSum <= 3.1 AND ClippingRankSum >= −3.1. An additional masking was applied, based on the all-sites base quality information output by GATK HaplotypeCaller (RRID:SCR_001876, v3.4–0, *DePristo et al., 2011*): DP >= 5*SOMY, DP <= 1.75* (chromosome median read depth) and GQ >= 10. Resulting samples were combined and SNPs with all reference or missing genotypes were removed.

## Phylogenetic reconstruction

For phylogenetic reconstruction from whole-genome polymorphism data, all 395,602 SNPs that are polymorphic within the species complex and have a maximum fraction of 0.2 non-called sites across all 151 samples were considered. Nei's distances were calculated for bi-allelic sites per chromosome with the R package StAMPP (v1.5.1, *Pembleton et al., 2013*), which takes into account aneuploidy across samples. Resulting distances matrices of Nei's distances per chromosome were weighted by chromosomal SNP count forming a consensus distance matrix, that was used for phylogenetic reconstruction with the Neighbor-Joining algorithm implemented in the R package APE (RRID:SCR_017343, v5.2, *Saitou and Nei, 1987*). For rooting of the tree, the phylogenetic reconstruction was repeated using three additional outgroup samples, of *L. major* (LmjFried, ENA: ERS001834; *Rogers et al., 2011*), *L. tropica* (P283, ENA: ERS218438; *Iantorno et al., 2017*) and *L. mexicana* (LmexU1103 v1, ENA: ERS003040; *Rogers et al., 2011*) (https://www.ebi.ac.uk/ena) using a total of 1,673,461 SNPs. Bootstrap replicates were generated by calculating distances matrices of Nei's distances for 10 kb windows and randomly sampling windows with replacement for a total of 1000 bootstrap replicates. For each bootstrap-replicate Neis' distances were summed up across windows, trees were generated with neighbour-joining and bootstrap support was provided for major branching nodes.

## Phylogenetic reconstruction of the *L. infantum* Linf1 group including additional brazilian isolates

Sequence reads of all 47 samples from the Linf1 group and of the 26 samples additional *L. infantum* strains isolated from human infections in Brazil (*Carnielli et al., 2018*) were trimmed with Trimmomatic (RRID:SCR_011848, v0.39, *Bolger et al., 2014*) including removal of paired-end adaptors using the options: 'ILLUMINACLIP:PEadaptors.fa:2:30:10 TRAILING:15 SLIDINGWINDOW:4:15 MIN-LEN:50'. Trimmed reads were mapped using BWA (RRID:SCR_010910, v0.7.17, *Li and Durbin, 2009*) using the bwa mem -M option. SNPs were called using GATK (RRID:SCR_001876, v4.1.2.0, *DePristo et al., 2011*): First, g.vcf files were generated for individual samples with the 'HaplotypeCaller' and parameters '-ERC GVCF --annotate-with-num-discovered-alleles --sample-ploidy 2'. Then individual g.vcf files were combined using 'GenomicsDBImport', SNPs across all samples were called using 'GenotypeGVCFs' and hard filtered using parameters "QD <2.0, MQ <50.0, FS >20.0, SOR > 2.5, BaseQRankSum < −3.1, ClippingRankSum < −3.1, MQRankSum <−3.1, ReadPosRankSum <−3.1 and DP <6'. The resulting vcf file were analysed in R: only SNPs with a missing fraction across samples < 0.2 were retained; Nei's distances between samples were called using the R package StAMPP (v1.5.1, *Pembleton et al., 2013*) and phylogenetic trees calculated with neighbour joining with the r package ape (RRID:SCR_017343, v5.3, *Paradis et al., 2004*).

## Phylogenetic reconstruction of maxicircles

Sequence reads were mapped against the maxicircle DNA of the reference strain, LV9 (MHOM/ET/1967/HU3), of *L. donovani* (TriTrypDB v46, RRID:SCR_007043) with SMALT (RRID:SCR_005498, v0.7.4, *Ponstingl, 2010*) using parameters: '-x -y 0.8 r −1 -i 1500' and duplicates were marked with picard, 'MarkDuplicates' (RRID:SCR_006525, v1.92, https://broadinstitute.github.io/picard/). Local indel realignments were performed on the resulting alignments with GATK using the

'RealignerTargetCreator' and 'IndelRealigner' with default settings (RRID:SCR_001876, v3.4–0, *DePristo et al., 2011*) and subsequently filtered for a mapping quality of 20 and proper pairs using samtools, parameters '-q 20 f 0 × 0002 F 0 × 0004 F 0 × 0008' (RRID:SCR_002105, v1.3, *Li et al., 2009*) SNP and Indel variants were called, hard filtered, selected and transformed to fasta sequences using GATK tools HaplotypeCaller, VariantFiltration, and FastaAlternateReferenceMaker (RRID:SCR_001876, v3.4–0, *DePristo et al., 2011*). Used parameters include: '–sample_ploidy 1 -dt NONE –annotateNDA' (HaplotypeCaller), 'QD <2.0, MQ <40.0, FS >13.0, SOR > 4, BaseQRankSum > 3.1 || BaseQRankSum < −3.1', ClippingRankSum > 3.1 || ClippingRankSum < −3.1, MQRankSum >3.1 || MQRankSum <−3.1, ReadPosRankSum >3.1 || ReadPosRankSum <−3.1, DP > \$DPmax, DP < \$DPmin (SNP, VariantFiltration), 'QD <2.0 || FS >200.0 || ReadPosRankSum <−20.0' (Indel, VariantFiltration) and '-IUPAC 1' (FastaAlternateReferenceMaker). We determined maxicircle coverage of individual isolates using samtools depth (RRID:SCR_002105, v1.3, *Li et al., 2009*). Not all samples contained sufficient maxicircle DNA (likely depending on the DNA extraction protocol used) (*Figure 4—figure supplement 4A*). We therefore only used samples that had a medium coverage of at least 20, resulting in 116 samples (*Figure 4—figure supplement 3* and *4*, *Supplementary file 3*) for subsequent analysis. As in the repetitive region of the maxicircle high quality mapping was not present, we assessed the minimum coverage across all 116 'good coverage' samples and based on that chose a region with a minimum coverage across those samples >= 10 for subsequent alignment and phylogenetic reconstruction (positions 984 to 17,162, *Figure 4—figure supplement 4B*). Resulting fasta sequences of individual maxicircles per isolates were aligned using MUSCLE (RRID:SCR_011812, v3.8.31, *Edgar, 2004*) with default parameter settings and the phylogeny was reconstructed with RaxML (RRID:SCR_006086, v7.0.3, *Stamatakis, 2006*) using parameters: 'raxmlHPC -f a -m GTRGAMMA -p 12345 -x 12345 -# 100'.

## Gene-feature annotation and GO enrichment analysis

All SNPs were annotated with gene features using the software SNPeff (RRID:SCR_005191, v4.2, *Cingolani et al., 2012*). Annotations for the reference genome *L. infantum*, JPCM5, were downloaded from TriTrypDB (v38, RRID:SCR_007043; *Aslett et al., 2010*). Several gene sets of interest were subsequently tested for Gene ontology (GO, RRID:SCR_002811) term enrichments for the ontology 'biological process'. GO mappings for *L. infantum* genes were downloaded from TriTrypDB (v38, RRID:SCR_007043), where 4704 of the 8299 annotated coding genes were also associated with a GO term. Enrichment of functional categories was tested using the weightFisher algorithm in topGO (RRID:SCR_014798, v2.34.0, *Alexa et al., 2006*) sing all genes annotated in the 'gene to GO' mapping file (v38). GO categories enriched with a p-value<0.05 (test: weightFish) were subsequently visualised with Revigo (RRID:SCR_005825, http://revigo.irb.hr/, assessed: February 2019, *Supek et al., 2011*) using default settings and rectangle sizes normalized by absolute p-value.

## Population structure and IBD analysis

To run ADMIXTURE (RRID:SCR_001263, v1.23, *Alexander et al., 2009*), SNP genotype calls were collapsed from polysomic to disomic for all chromosomes and only biallelic SNPs were included. SNPs were filtered and thinned, removing SNPs with copies of the minor allele in less than four samples and one of two neighbouring SNPs with a minimum distance <250 bp. Using a five-fold cross-validation (CV) the optimal values of $K$ (smallest CV error) was determined to be 8 and 11 but we also explored different $K$ values. The value of $K$ chosen was robust to different CV schemes. For IBD analysis, we calculated correlations between genetic and geographical pairwise distances between isolated strains using the Mantel test (R package ade4, v1.7–13, *Dray and Dufour, 2007*). Genetic distances were estimated as Neis' D based on genome-wide SNP information using the R package StAMPP (v1.5.1, *Pembleton et al., 2013*). Geographic distances were calculated as geodesic distances between the respective countries of sample origin using the R package lmap (v1.32).

## Haplotype-based analysis of hybridisation in CUK isolates

We used SNP calls across all the original 12 CUK isolates from *Rogers et al. (2014)* and called fractions of heterozygous alleles and homozygous differences from the JPCM5 reference for 5 kb windows for each isolate. Mean heterozygous and homozygous fractions per window were calculated as genomic regions with either no SNP or increased number of homozygous differences (see also

*Rogers et al., 2014*). Putative parent blocks were identified using consecutive windows with mean heterozygous fractions < 0.0002 (1 SNP/5 kb) and mean homozygous fractions either <0.0004 (2 SNP/5 kb) for the JPCM5-like parent or >0.001 (5 SNP/5 kb) for the unknown parent. Those thresholds are quite stringent (*Figure 4—figure supplement 7*), but allowed conservative calling of putative parental haplotype regions. For each parent, we selected the largest four regions conditioning on at most one block per chromosome (resulting block sizes from 150 to 215 kb; *Figure 4—figure supplement 7*). Phylogenetic trees for each of the eight regions were then reconstructed based on polyploid genotypes of all 151 isolates and three outgroups (LmjFried, *L. major*, ENA: ERS001834; P283, *L. tropica*, ENA: ERS218438; LmexU1103 v1, *L. mexicana*, ENA: ERS003040; https://www.ebi. ac.uk/ena) using Nei's distances calculated with StAMPP (v1.5.1, *Pembleton et al., 2013*) and the neighbour joining algorithm (R package ape, v5.2) in R (*Supek et al., 2011*).

## Population genomics characterisation of the groups

For the population genomics characterization of the largest groups identified based on the global phylogeny (*Figure 1A*), isolates that were identified as putative mixtures of clones were removed. These were BPK157A1 (Ldon1), GILANI (Ldon3), LRC-L53 (Ldon5) and Inf152 (Linf1) and their respective groups are indicated by an asterisk (*). Polyploid genotype calls were transformed into diploid calls by transforming multiploid heterozygous sites into diploid heterozygous sites and polyploid homozygotes into diploid homozygotes. Linkage disequilibrium for each group was then calculated as genotype correlations of the transformed diploid calls using vcftools (RRID:SCR_001235, v0.1.14, parameter: `--geno-r2`) (*Danecek et al., 2011*). For each group LD was calculated including all available samples in a group. For groups containing more than seven samples, three 'pseudo-replicates' were generated by random sampling without replacement. This way results were comparable between groups and the smallest groups containing only seven samples. $F_{ST}$ between all group pairs was calculated for polymorphic sites with a minimum fraction of 0.8 called sites across all 151 samples as described in 'Phylogenetic reconstruction' using the R package StAMPP (v1.5.1, *Pembleton et al., 2013*).

## Genomic characterisation of individual isolates

Within isolate genome-wide heterozygosity was calculated using the formula:

$$1 - \frac{1}{m}\sum_{j=1}^{m}\sum_{i=1}^{k_j} pij^2$$

where $p_i$ is the frequency of the $i^{th}$ of k alleles for a given SNP genotype and the $1^{st}$ summation sums over all $m$ SNP loci for a given isolate. Here, genotype calls consider the correct somy for each isolate and chromosome as described above (see 'Variant calling'). Isolate specific allele frequency spectra were obtained using mapped bam files including duplicate identification and indel realignment as described above (see 'Read Mapping Pipeline'). Bam files were subsequently filtered using samtools view (RRID:SCR_002105, v1.3, *Li et al., 2009*) to only keep reads mapped in a proper pair with mapping quality of at least 20. Filtered bam files were summarised using samtools mpileup (RRID:SCR_002105, v1.3, *Li et al., 2009*) with arguments -d 3500 -B -Q 10 limiting the per sample coverage to 3500, disabling probabilistic realignment for the computation of base alignment quality and a minimum base quality of 10. The resulting mpileup file was converted to sync format summarising SNP allele counts per isolate using the mpileup2sync.jar script requiring a minimum base quality of 20 (*Kofler et al., 2011*). For the 11 samples with extreme allele frequency spectra, heterozygous SNPs were additionally filtered for the highest SNP calling quality of 99 (~$10^{-10}$ probability of an incorrect genotype) and alternate alleles that were called as homozygous alternate alleles in at least five other isolates to confirm the presence of the skewed allele frequency spectra (*Figure 4—figure supplement 11*).

## Copy number variation

To identify large copy number variants (CNVs), realigned bam files for each sample were filtered for proper-pairs and PCR or optical duplicates were removed using samtools view (RRID:SCR_002105, v1.3, *Li et al., 2009*). Coverage was then determined using bedtools genomecov (RRID:SCR_006646, v2.17.0) with parameters: '-d -split' (*Quinlan and Hall, 2010*). Large duplications and

deletion were identified using custom scripts in R (*R Development Core Team, 2013*): genome coverage was determined for 5 kb non-overlapping windows along the genome and each window was normalized by the haploid chromosome coverage of the respective chromosome and sample (i.e. median chromosome coverage divided by somy of the respective chromosome and sample). Large CNVs were identified through stretches of consecutive windows with a somy-normalized median coverage >= 0.5 or<=−0.5 for duplications and deletions, respectively, a minimum length of 25 kb and a median normalized coverage difference across windows >= 0.9 (*Supplementary file 6*). To identify large CNVs across samples at identical positions and variant type, we grouped CNVs across samples with identical start and end positions within <= 10 kb (i.e. up to two 5 kb windows difference) (*Supplementary file 7*). CNVs of individual genes were determined based on the filtered bam files (see genome coverages) with bedtools coverage (RRID:SCR_006646, v2.17.0) using parameters '-d -split' (*Quinlan and Hall, 2010*) and analysing gene coverages in R (*R Development Core Team, 2013*). The coverage of each gene was approximated by its median coverage and normalized by the haploid coverage of the respective chromosome and sample (*Supplementary file 9*).

## Identification of repeated sequences in the reference genome of *L. infantum*

Repeated sequences in the JPCM5 *L. infantum* reference had previously been identified for assembly v3 (GeneDB, RRID:SCR_002774) in *Ubeda et al. (2014)*. We obtained the respective reference sequence from the author as v3 was no longer available on GeneDB. Repeated sequences were extracted based on this reference and positional information from *Ubeda et al. (2014)* with bedtools getfasta -s (RRID:SCR_006646, v2.29.0, *Quinlan and Hall, 2010*). Locations of the extracted repeat sequences in the reference genome JPCM5 (TriTrypDB v38, RRID:SCR_007043; *Aslett et al., 2010*) were identified with nucmer using default parameters (*Marçais et al., 2018*). 100% matches of the repeats in the new reference genome were annotated with the respective RAG number (*Ubeda et al., 2014*). A comparison of the previously used reference genome used for repeat identification in *Ubeda et al. (2014)* and version v38 (TriTryDB) with nucmer (*Marçais et al., 2018*) further showed a missing region on chromosome 27 of 269,698 bp in the previous genome version corresponding to positions 199,468–269,164 in v38 (*Figure 7—figure supplement 8A*). As this region contained a deletion of interest on chromosome 27 (*Figure 7A*, *Figure 7—figure supplement 7A*) common to a subset of our strains, we also screened for unknown repeats in the respective region using nucmer with parameters '--maxmatch --nosimplify --mincluster 30 --minmatch 7' within the region: LinJ.27:190000–300000 in the reference genome TriTrypDB v38 (*Figure 7—figure supplement 8B*, *Supplementary file 13*).

## Measures of selection

For all genes with annotated mRNAs in TriTrypDB (v38, RRID:SCR_007043; *Aslett et al., 2010*), the longest open reading frames (ORF) were identified using a custom python script, resulting in 8234 genes with and five without ORFs. ORFs were then edited for SNP variation in both species using custom python scripts. Numbers of polymorphic differences within a species versus fixed differences to an outgroup of both, non-synonymous and synonymous sites, were annotated and tested for significance with Fisher's exact test using previously implemented software (*Holloway et al., 2007*). This was done for each gene and species always using the respective other species as an outgroup and removing sites polymorphic in the outgroup. An unbiased version of the α statistic (*Smith and Eyre-Walker, 2002*; *Stoletzki and Eyre-Walker, 2011*), intended to estimate the proportion of non-synonymous substitutions fixed by positive selection across genes, was calculated with a custom R script.

## Data availability

The 97 samples sequenced for this study are deposited in ENA under the study accession numbers: PRJEB2600 (ERP000767), PRJEB2724 (ERP000966), PRJEB8947 (ERP009989) and PRJEB2115 (ERP000169) (https://www.ebi.ac.uk/ena/data/view). All metadata on the 151 isolates including ENA accession numbers of individual samples are summarized in *Supplementary file 1* (see also https://microreact.org/project/_FWlYSTGf; *Argimón et al., 2016*). Summary statistics and annotations from this study are available in *Supplementary files 1*, *2*, *3*, *4*, *5*, *6*, *7*, *8*, *9*, *10*, *11*, *12* and *13*. Analysis

scripts generated and used in this study along with the corresponding data files are available on github https://github.com/susefranssen/Global_genome_diversity_Ldonovani_complex. (*Franssen and Cotton, 2020*; copy archived at https://github.com/elifesciences-publications/Global_genome_diversity_Ldonovani_complex).

## Acknowledgements

We are very grateful to Gad Baneth from the Koret School of Veterinary Medicine, Hebrew University of Jerusalem, Israel; Patrick Bastien from the Centre National de Référence des Leishmanioses Montpellier, France; Sayda Hassan El Safi from Khartoum University, Khartoum, Sudan; Olga Francino Martí from Universitat Auònoma de Barcelona, Spain; Marina Gramiccia from the Infectious Diseases Department Istituto Superiore di Sanità, Rome, Italy; Ketty Soteriadou and Evi Gouzelou from the Hellenic Pasteur Institute Athens, Greece; A.K.M. Shamsuzzaman from Mymensingh Medical College and Be-Nazir Ahmed from Institute of Epidemiology, Disease Control and Research, Bangladesh; Vanessa Yardley from the London School of Hygiene and Tropical Medicine and Peter Walden from Charité Universitätsmedizin Berlin, Germany, each for providing isolates for this study. We also thank the *Leishmania* Collection of the Oswaldo Cruz Foundation, Brazil and Elisa Cupolillo for providing the isolates MHOM/BR/2003/MAM, MHOM/BR/2007/ARL, MHOM/BR/2007/WC, voucher numbers: IOC/L2651, IOC/L2935 and IOC/L3015, respectively. Further details of collaborators and their donated samples are given in *Supplementary file 1*.

This project was supported by Wellcome through its core funding of the Wellcome Sanger Institute (grants WT098051 and WT206194) and by the EU framework program FP7- 222895.

## Additional information

### Funding

| Funder | Grant reference number | Author |
|---|---|---|
| Wellcome Trust | Wellcome Sanger Institute core funding, WT098051 | Susanne U Franssen<br>Caroline Durrant<br>Mandy J Sanders<br>Matthew Berriman<br>James A Cotton |
| Wellcome Trust | Wellcome Sanger Institute core funding, WT206194 | Susanne U Franssen<br>Caroline Durrant<br>Mandy J Sanders<br>Matthew Berriman<br>James A Cotton |
| EU Framework Programme for Research and Innovation | FP7- 222895 | Tim Downing<br>Hideo Imamura<br>Olivia Stark<br>Jean-Claude Dujardin<br>Matthew Berriman<br>Gabriele Schönian<br>James A Cotton |

The funders had no role in study design, data collection and interpretation, or the decision to submit the work for publication.

### Author contributions

Susanne U Franssen, Conceptualization, Data curation, Formal analysis, Validation, Investigation, Visualization, Methodology, Writing - original draft, Project administration; Caroline Durrant, Data curation, Formal analysis, Methodology, Writing - review and editing; Olivia Stark, Conceptualization, Resources, Data curation, Writing - review and editing; Bettina Moser, Resources, Data curation, Formal analysis, Writing - review and editing; Tim Downing, Conceptualization, Data curation, Formal analysis, Writing - review and editing; Hideo Imamura, Data curation, Formal analysis, Writing - review and editing; Jean-Claude Dujardin, Matthew Berriman, Gabriele Schönian, Conceptualization, Resources, Supervision, Funding acquisition, Writing - review and editing; Mandy J Sanders, Data curation, Project administration, Writing - review and editing; Isabel Mauricio, Michael A Miles,

Lionel F Schnur, Charles L Jaffe, Abdelmajeed Nasereddin, Henk Schallig, Matthew Yeo, Tapan Bhattacharyya, Mohammad Z Alam, Resources, Writing - review and editing; Thierry Wirth, Conceptualization, Writing - review and editing; James A Cotton, Conceptualization, Formal analysis, Supervision, Funding acquisition, Validation, Visualization, Writing - original draft, Project administration

### Author ORCIDs

Susanne U Franssen 🔟 https://orcid.org/0000-0002-4725-1793
Tim Downing 🔟 http://orcid.org/0000-0002-8385-6730
Hideo Imamura 🔟 http://orcid.org/0000-0002-3282-1319
Jean-Claude Dujardin 🔟 http://orcid.org/0000-0002-9217-5240
Abdelmajeed Nasereddin 🔟 http://orcid.org/0000-0001-8162-420X
Matthew Berriman 🔟 http://orcid.org/0000-0002-9581-0377
James A Cotton 🔟 https://orcid.org/0000-0001-5475-3583

### Ethics

Human subjects: Only previously collected parasite isolates from humans and animals have been used in this study. The parasites from human cases had been isolated as part of normal diagnosis and treatment with no unnecessary invasive procedures and data on human isolates were encoded to maintain anonymity.

### Decision letter and Author response

Decision letter https://doi.org/10.7554/eLife.51243.sa1
Author response https://doi.org/10.7554/eLife.51243.sa2

## Additional files

### Supplementary files

• Supplementary file 1. Metadata on all 151 isolates analysed based on whole genome sequencing data.

• Supplementary file 2. IBD analysis. Summary of correlations between genetic and geographical distances between sample-pairs from different groups using the Mantel test. Genetic distances were measured as Nei's distances for genome-wide SNPs and geographical distances were calculated as geodesic distances between countries of sample origin.

• Supplementary file 3. Samples with identical maxicircle sequences.

• Supplementary file 4. Marker SNPs for each individual group and between species group, excluding samples of putatively mixed clone origin, annotated with SNPeff.

• Supplementary file 5. SNPs segregating in at least 5 of the eight identified groups with respective group frequencies and SNPeff annotations.

• Supplementary file 6. Metadata on large indels (>=25 kb) called for each of the 151 samples.

• Supplementary file 7. Metadata of the 245 unique (by genomic position) large indels (>=25 kb) called across all 151 samples.

• Supplementary file 8. Localisation of the LD1/CD1 locus in the reference assembly of LinJPCM5, TriTrypDB v38. Sequence matches were identified with nucmer (default parameters) and only the longest match for each query sequence is reported. (spos/epos are start and end positions of the sequence matches seq1 and seq2.)

• Supplementary file 9. Meta data on gene copy number across all 8,330 genes and all 151 samples.

• Supplementary file 10. Summary of gene copy number analysis including genes with frequent copy number changes and their functional enrichment using GO term enrichment analysis. (A) List of all genes with median copy number change across all 151 samples unequal to 0. (B) Enriched GO terms across genes that show a median copy number decrease (<=1). (C) Enriched GO terms across genes that show a median copy number increase (>=1). (D) Enriched GO terms across genes that show a

median copy number increase (>=4). (E) Listed are genes that contribute to the GO enrichment of genes with a median copy number change <= 1,>=1 and>=4 across all samples, respectively.

• Supplementary file 11. Metadata on genes in *L. donovani* and *L. infantum* with a McDonald-Kreitman test p-value<0.05.

• Supplementary file 12. GO term enrichment for genes including marker SNPs for each of the eight identified groups and between *L. infantum* and *L. donovani* samples that show predicted effects of moderate to high effect (SNPeff, *Cingolani et al., 2012*).

• Supplementary file 13. Repeated sequences on chromosome 27, region 190,000–300,000, reference genome *L. infantum*, JPCM5, v38 TriTrypDB identified with nucmer (self comparison, parameters –maxmatch –nosimplify –mincluster 30 –minmatch 7).

• Transparent reporting form

## Data availability

The 97 samples sequenced for this study are deposited in ENA under the study accession numbers: PRJEB2600 (ERP000767), PRJEB2724 (ERP000966), PRJEB8947 (ERP009989) and PRJEB2115 (ERP000169) (https://www.ebi.ac.uk/ena/data/view). Full accession details/sample accessions of these 97 samples are: PRJEB2600 (ERS104335, ERS104333, ERS104323, ERS218540, ERS177300, ERS177299, ERS177296, ERS177295, ERS218539, ERS082780, ERS097150, ERS097157, ERS082781, ERS097158, ERS082784, ERS3773247, ERS097154, ERS104327, ERS104322, ERS104329, ERS104330, ERS177293, ERS177294, ERS3773245, ERS104316, ERS104318, ERS097142, ERS3773246, ERS3773248, ERS104315, ERS097138, ERS066256, ERS082776, ERS104324, ERS082775, ERS082777, ERS066257, ERS082774, ERS066262, ERS082773, ERS3773249, ERS3773250, ERS066261, ERS3773251, ERS104312, ERS097135, ERS104323, ERS040396, ERS3773252, ERS3773254, ERS3773253, ERS3773255, ERS3773256, ERS097153, ERS104320, ERS066265, ERS082783, ERS082782, ERS104313, ERS066259, ERS097141, ERS097136, ERS104325, ERS097143, ERS097148, ERS3773257, ERS3773258, ERS3773259, ERS066258, ERS040394, ERS097156, ERS097145, ERS066264, ERS3773261, ERS066260, ERS097140, ERS066263, ERS082779, ERS097147, ERS3773263, ERS3773260, ERS3773262, ERS104314, ERS097155, ERS3773264, ERS097139, ERS3773244, ERS040393, ERS407440), PRJEB2724 (ERS100733, ERS419988, ERS419990, ERS419987, ERS419989, ERS419991), PRJEB8947 (ERS009628, ERS008275), PRJEB2115 (ERS001888). All metadata of the 151 isolates we analysed in depth including another 54 sequenced samples from other studies are summarized in Supplementary file 1 also detailing the respective ENA accession numbers of individual samples (see also https://microreact.org/project/_FWlYSTGf; Argimón et al., 2016). Summary statistics and annotations from this study are available in Supplementary files 1–13. Analysis scripts generated and used in this study along with the corresponding data files are available on github https://github.com/susefranssen/Global_genome_diversity_Ldonovani_complex (copy archived at https://github.com/elifesciences-publications/Global_genome_diversity_Ldonovani_complex).

The following datasets were generated:

| Author(s) | Year | Dataset title | Dataset URL | Database and Identifier |
|---|---|---|---|---|
| Franssen SU, Durrant C, Stark O, Moser B, Downing T, Imamura H, Dujardin J-C, Sanders M, Mauricio I, Miles MA, Schnur LF, Jaffe CL, Nasereddin A, Schallig H, Yeo M, Bhattacharyya T, Alam MZ, Berriman M, Wirth T, Schönian G, Cotton JA | 2019 | Global genome diversity of the *Leishmania donovani* complex | https://www.ebi.ac.uk/ena/data/view/PRJEB2600 | EBI European Nucleotide Archive, PRJEB2600 |
| Franssen SU, Durrant C, Stark O, | 2019 | Global genome diversity of the *Leishmania donovani* complex | https://www.ebi.ac.uk/ena/data/view/ | EBI European Nucleotide Archive, |

| Author(s) | Year | Dataset title | Dataset URL | Database and Identifier |
|---|---|---|---|---|
| Moser B, Downing T, Imamura H, Dujardin J-C, Sanders M, Mauricio I, Miles MA, Schnur LF, Jaffe CL, Nasereddin A, Schallig H, Yeo M, Bhattacharyya T, Alam MZ, Berriman M, Wirth T, Schönian G, Cotton JA | | | PRJEB2724 | PRJEB2724 |
| Franssen SU, Durrant C, Stark O, Moser B, Downing T, Imamura H, Dujardin J-C, Sanders M, Mauricio I, Miles MA, Schnur LF, Jaffe CL, Nasereddin A, Schallig H, Yeo M, Bhattacharyya T, Alam MZ, Berriman M, Wirth T, Schönian G, Cotton JA | 2019 | Global genome diversity of the *Leishmania donovani* complex | https://www.ebi.ac.uk/ena/data/view/PRJEB8947 | EBI European Nucleotide Archive, PRJEB8947 |
| Franssen SU, Durrant C, Stark O, Moser B, Downing T, Imamura H, Dujardin J-C, Sanders M, Mauricio I, Miles MA, Schnur LF, Jaffe CL, Nasereddin A, Schallig H, Yeo M, Bhattacharyya T, Alam MZ, Berriman M, Wirth T, Schönian G, Cotton JA | 2019 | Global genome diversity of the *Leishmania donovani* complex | https://www.ebi.ac.uk/ena/data/view/ERS001888 | EBI European Nucleotide Archive, ERS001888 |

The following previously published datasets were used:

| Author(s) | Year | Dataset title | Dataset URL | Database and Identifier |
|---|---|---|---|---|
| Imamura H, Downing T, VandenBroeck F, Sanders MJ, Rijal S, Sundar S, Mannaert A, Vanaerschot M, Berg M, DeMuylder G, Dumetz F, Cuypers B, Maes I, Domagalska M, Decuypere S, Rai K, Uranw S, Bhattarai NR, Khanal B, Prajapati VK, Sharma S, Stark O, Schönian G, De Koning HP, Settimo L, Vanhollebeke B, Roy S, Ostyn B, Boelaert M, Maes L, Berriman M, Dujardin J-C, Cotton JA | 2016 | Evolutionary genomics of epidemic visceral leishmaniasis in the Indian subcontinent | https://www.ebi.ac.uk/ena/data/view/ERP000140 | EBI European Nucleotide Archive, ERP000140 |
| Rogers MB, Downing T, Smith BA, Imamura H, Sanders M, Svobodova | 2014 | Genomic Confirmation of Hybridisation and Recent Inbreeding in a Vector-Isolated *Leishmania* Population | https://www.ebi.ac.uk/ena/data/view/PRJEB2473 | EBI European Nucleotide Archive, PRJEB2473 |

| | | | | |
|---|---|---|---|---|
| M, Volf P, Berriman M, Cotton JA, Smith DF | | | | |
| Zackay A, Cotton JA, Sanders M, Hailu A, Nasereddin A, Warburg A, Jaffe CL | 2018 | Genome wide comparison of Ethiopian *Leishmania donovani* strains reveals differences potentially related to parasite survival | https://www.ebi.ac.uk/ena/data/view/PRJEB14372 | EBI European Nucleotide Archive, PRJEB14372 |
| Zhang WW, Ramasamy G, McCall L-I, Haydock A, Ranasinghe S, Abeygunasekara P, Sirimanna G, Wickremasinghe R, Myler P, Matlashewski G | 2014 | Genetic analysis of *Leishmania donovani* tropism using a naturally attenuated cutaneous strain | https://www.ncbi.nlm.nih.gov/bioproject/210295 | NCBI BioProject, SRS484824 |
| Peacock CS, Seeger K, Harris D, Murphy L, Ruiz JC, Quail MA, Peters N, Adlem E, Tivey A, Aslett M, Kerhornou A, Ivens A, Fraser A, Rajandream M-A, Carver T, Norbertczak H, Chillingworth T, Hance Z, Jagels K, Moule S, Ormond D, Rutter S, Squares R, Whitehead S, Rabbinowitsch E, Arrowsmith C, White B, Thurston S, Bringaud F, Baldauf SL, Faulconbridge A, Jeffares D, Depledge DP, Oyola SO, Hilley JD, Brito LO, Tosi LRO, Barrell B, Cruz AK, Mottram JC, Smith DF, Berriman M | 2007 | Comparative genomic analysis of three *Leishmania* species that cause diverse human disease | https://www.ebi.ac.uk/ena/data/view/ERS001832 | EBI European Nucleotide Archive, ERS001832 |
| Rogers MB, Hilley JD, Dickens NJ, Wilkes J, Bates PA, Depledge DP, Harris D, Her Y, Herzyk P, Imamura H, Otto TD, Sanders M, Seeger K, Dujardin J-C, Berriman M, Smith DF, Hertz-Fowler C, Mottram JC | 2011 | Chromosome and gene copy number variation allow major structural change between species and strains of *Leishmania* | https://www.ebi.ac.uk/ena/data/view/PRJEB2115 | EBI European Nucleotide Archive, PRJEB2115 |
| Iantorno SA, Durrant C, Khan A, Sanders MJ, Beverley SM, Warren WC, Berriman M, Sacks DL, Cotton JA, Grigg ME | 2017 | Gene Expression in *Leishmania* Is Regulated Predominantly by Gene Dosage | https://www.ebi.ac.uk/ena/data/view/ERS218438 | EBI European Nucleotide Archive, ERS218438 |
| Carnielli JBT, Crouch K, Forrester S, Silva VC, Carvalho SFG, Damasceno JD, Brown E, Dickens NJ, Costa | 2018 | A *Leishmania infantum* genetic marker associated with miltefosine treatment failure for visceral leishmaniasis | https://www.ncbi.nlm.nih.gov/bioproject/PRJNA494801 | NCBI BioProject, PRJNA494801 |

DL, Costa CHN,
Dietze R, Jeffares
DC, Mottram JC

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
