## [Decision Letter]

**Acceptance summary:**

This is an extremely thorough and careful analysis of genetic variation in strains from the *Leishmania donovani* complex, especially of non-clonal infections. Overall, this is a masterclass in analysis methodology. Importantly, only one subgroup of the complex shows strong geographical structure. Similarly, only some populations are genetically diverse and show strong evidence for hybridization and recombination, while others do not. Of particular interest is that there is nevertheless shared variation between the two subgroups.

**Decision letter after peer review:**

Thank you for submitting your article "Global genome diversity of the *Leishmania donovani* complex" for consideration by *eLife*. Your article has been reviewed by two peer reviewers, and the evaluation has been overseen by a Reviewing Editor and Detlef Weigel as the Senior Editor. The following individuals involved in review of your submission have agreed to reveal their identity: Ouellette Marc (Reviewer #1).

The reviewers have discussed the reviews with one another and the Reviewing Editor has drafted this decision to help you prepare a revised submission.

Both reviews are included below. In addition to addressing the requests from the reviewers, we suggest that – without going beyond what can be deduced from the data, of course – you pay a little more attention to isolation-by-distance, and if possible make inferences about geographic movements, recent population sizes etc.

The scripts that you used should be uploaded unto GitHub, and the reference supplied.

Reviewer #1:

Franssen et al., have used NGS to look at the diversity of the *Leishmania donovani* complex. They sequenced 97 novel strains and reanalysed the genome of 54. The generation of the sequencing data was relatively easy but its analysis was not. Several co-authors have contributed extensively already to our understanding of genome diversity in *Leishmania* and as such this VERY extensive analysis is first rated. The main conclusion is that *L. donovani* is more heterogeneous than expected and followed geography while *L. infantum* seems less diverse. Below are comments to be taken into account by the authors.

1) While acknowledged in the Discussion the numbers of *L. infantum* isolates from South America (and North Africa for that matter) are low. This should be acknowledged earlier. The Turkey/Cyprus arm of *L. infantum* already shows the possibility for heterogeneity with *L. infantum*. Similarly for a zoonotic parasite, the animal isolates are also low. Are they infantum isolates from man and animals from the same region in this study?

2) In their CNV analysis they found a 25 kb deletion on chromosome 27. From the GO term have they check whether those cells have altered motility? This should be relatively simple.

3) In the same CNV analysis they found an increase CNV on chromosome 35. It coincides, at least in part, with the CD1/LD1 locus that Ken Stuart and others have characterized 25-30 years ago. If I recall correctly this has been observed in many species. This should be discussed. Notably, this region encodes for the biopterin transporter BT1.

4) Deletion and duplication in *Leishmania* would normally occur at the level of repeated sequences. Was this analysed for chromosome 27/35 CNV?

5) In the analysis of drug resistance markers for AQP1 I would like to make two points. One this is on the polyploid chromosome 31 and second it is next to a telomere. Can this have an impact on the data presented or its interpretation?

6) The authors described at great length linkage disequilibrium but at the end I am unclear whether there is evidence in their analysis for sexual recombination. Can they be more explicit?

Reviewer #2:

Franssen et al. examine illumina genome sequences from 151 isolates of *Leishmania donovani* (*L.d.*) and *L. infantum* (*L.i*.), causative agents of visceral leishmaniasis, from across most of the range of these parasites. The dataset includes 97 new genome sequences with the remaining samples from published datasets. The analysis documents strong differentiation between *L.d.* and *L.i.*, differences in the population structure of these two species, variable levels of diversity and recombination within populations and hybrid origin of some genotypes, extensive mosaic aneuploidy and copy number variation, as well as providing evidence for links between CNVs and drug resistance and signature of selection that may be related to pathogenesis/parasite burden.

Strong features

This manuscript provides an extremely thorough and careful analysis of genetic variation in *L.d.* and *L.i.*. I was impressed with by the careful examination of the data to differentiate between heterozygosity and non-clonal infections, and by the careful analysis of LD decay with distance between markers, taking into account sample size and population structure. Overall, this is a masterclass in analysis methodology and certainly fulfills *eLife*'s criteria that "Articles must be methodologically and scientifically rigorous, ethically conducted, and objectively presented according to the appropriate community standards."

By examining samples from across the range of these species, the analysis clearly reveals fundamental differences between *L.d.* and *L.i.* and between different population samples of these parasites. While *L.d.* shows strong geographical structure, this is not the case for *L.i.* where closely related parasites are distributed across different continents. Similarly, some populations are genetically diverse and show strong evidence for hybridization and recombination (e.g. Turkey), other populations (e.g. Indian *L.d.*) show minimal diversity, and signatures of purely clonal propagation.

The analysis of copy number and ploidy levels is extremely thorough. I was particularly interested that 28% of CNVs were found in both species, suggesting that either these are old shared polymorphisms or that they have been selected independently in the two species. Resolving the breakpoints (perhaps by searching for paired end reads that cross chromosome breakage sites) could help differentiate between these alternatives and also provide some clues as to the sequence motifs involved in chromosome breakage and copy number change, and provide some novelty to this study (see notes below). I also like the hypothesis that rapid change in ploidy prevented accumulation of heterozygous sites during clonal division, although tests of this hypothesis were inconclusive.

Weak features

While the article is extremely rich in detail, I felt that it only incrementally increases our knowledge of leishmania population genomics, over previous work published by these and other authors. Previous work has demonstrated evidence for hybridization and recombination in both the laboratory and some field populations, has shown that Indian *L.d.* are extremely closely related, documented that mosiac aneuploidy is a central feature of leishmania, and identified strong association between copy number variants and drug resistance. I came away thinking that this is an impressive analysis, but wondering what were the key novel findings from this work – perhaps these could be more clearly highlighted in the Discussion?

I was unsure why only a small subset of the published sequences were included in this analysis. For example, 46 sequences are available from Brazil but were not included, while only a 6 of 41 available from Ethiopia were included. There may be good reason for omission of samples (data not made available, or failed read depth cutoffs etc.), but the manuscript would be improved by some rigid criteria for sample inclusion.

Overall, this is a comprehensive, carefully conducted analysis that adds valuable detail to our understanding of *Leishmania* population genomics.

---

## [Author Response]

Both reviews are included below. In addition to addressing the requests from the reviewers, we suggest that – without going beyond what can be deduced from the data, of course – you pay a little more attention to isolation-by-distance, and if possible make inferences about geographic movements, recent population sizes etc.We have now added analysis of isolation-by-distance and included the results in the section “Evolution of the L. donovani complex” and Supplementary file 2.

With respect to making quantitative inferences about population size and movement: This is challenging in the Leishmania context, where the non-standard genetic system of only occasional sexual reproduction and frequent aneuploidy variation means that standard models are not applicable. We think these aspects are best left until we have a better theoretical understanding of Leishmania genetics. Moreover, we think a more comprehensive sampling of some regions (e.g. the Middle East, Central Asia) is necessary before inferences on geographical movements can be robust and quantitative. However, we now point out and discuss geographical movements inferred by the observation of hybrids between putative “parental groups” from different geographical regions and introductions of the parasite into the New World.

The scripts that you used should be uploaded unto GitHub, and the reference supplied.

We have now loaded the scripts onto github. The link of the github page is provided in the “Data availability” section.

Reviewer #1:1) While acknowledged in the Discussion the numbers of L. infantum isolates from South America (and North Africa for that matter) are low. This should be acknowledged earlier.We now include an additional 26 L. infantum strains isolated from human infections in Brazil from (Carnielli et al., 2018) in our analysis to demonstrate the monophyly of Central/ and South American strains of the MON-1 zymodeme. These results are presented under section “Evolution of the L. donovani complex” and Figure 1—figure supplement 2B also acknowledging sparse sampling of North American *L. infantum* for our general analysis earlier in the manuscript (Material and methods).

The Turkey/Cyprus arm of L. infantum already shows the possibility for heterogeneity with L. infantum.We now already state this earlier.Similarly for a zoonotic parasite, the animal isolates are also low. Are they infantum isolates from man and animals from the same region in this study?We now refer to genetic relatedness in the context of host- and geographic origin. We show that within the “MON-1” clade, where we have almost all dog derived samples and which is known to be the predominant clade in dogs, parasite samples cluster by geographic origin rather than host (Figure 1—figure supplement 2A).

2) In their CNV analysis they found a 25 kb deletion on chromosome 27. From the GO term have they check whether those cells have altered motility? This should be relatively simple.

This is an interesting experiment to suggest, however, we feel this is beyond the scope of our analysis for several reasons:

1) As stated in Figure 7, the enrichment in the GO analysis is only due to a single gene annotated with “motility”, while in total 17 genes are present in the deleted region that might all equally be the functionally important “driver” of this deletion (all genes within that region: LinJ.27.2630, LinJ.27.2550, LinJ.27.2540, LinJ.27.2610, LinJ.27.2560, LinJ.27.2580, LinJ.27.2620, LinJ.27.2570, LinJ.27.2590, LinJ.27.2660, LinJ.27.2530, LinJ.27.2640, LinJ.27.2520, LinJ.27.2670, LinJ.27.2680, LinJ.27.2600, LinJ.27.2650). We have now made this clearer in the main text. In this context, picking this single target to experimentally validate seems rather arbitrary.

2) Samples for this project were assembled from a large number of collaborators/ co-authors and were obtained as DNA, so cryopreserved stabilates are not easily available to us and partly might not exist any longer for many of these isolates.

3) In the same CNV analysis they found an increase CNV on chromosome 35. It coincides, at least in part, with the CD1/LD1 locus that Ken Stuart and others have characterized 25-30 years ago. If I recall correctly this has been observed in many species. This should be discussed. Notably, this region encodes for the biopterin transporter BT1.We have now mapped the previously described CD1/LD1 locus and show its location with respect to the identified copy number variants (CNVs) and point it out in the Discussion (Figure 7, Figure 7—figure supplement 2, Supplementary file 8).

4) Deletion and duplication in Leishmania would normally occur at the level of repeated sequences. Was this analysed for chromosome 27/35 CNV?For the regions around CNVs on chromosomes 27 and 35, we have now added information of previously known (Ubeda et al., 2014) and newly identified repeat sequences. Surprisingly, these were not always associated with repeated sequences (Figure 7—figure supplement 7, 8, Supplementary file 13).

5) In the analysis of drug resistance markers for AQP1 I would like to make two points. One this is on the polyploid chromosome 31 and second it is next to a telomere. Can this have an impact on the data presented or its interpretation?

The gene copy number changes we report are calculated with respect to the individual chromosome dosage for each isolate, so the polyploidy of chromosome 31 is taken into account in the results.

With respect to the position of AQP1 proximate to the telomere: The AQP1 locus (LinJ.31.0030, LINF_310005100) is the second last annotated gene from the proximate telomere with only a methyltransferase (LinJ.31.0010, LINF_310005000) being closer to the telomere. Neither of the genes overlapped with the genomic regions we masked due to decreased mappability, suggesting high quality mappings within those two gene regions. Moreover, out of 52 samples with an increased copy number (CN) of AQP1, most (78%) did not show a CN change at LinJ.31.0010, and of 6 samples with decreased AQP1 CN 66% did not show a CN change at the second locus (data summarised from Supplementary file 9 but summary not presented in the manuscript). This indicates that CN changes were typically independent between both genes and consequently the observed CN change in the AQP1 gene should also typically be independent from amplifications of the telomere.Therefore, we do not see any particular impact of the AQP1 location on the presented results.

6) The authors described at great length linkage disequilibrium but at the end I am unclear whether there is evidence in their analysis for sexual recombination. Can they be more explicit?We are now more explicit about this in the relevant text sections (Results and Discussion).Reviewer #2:Strong featuresThis manuscript provides an extremely thorough and careful analysis of genetic variation in L.d. and L.i.. I was impressed with by the careful examination of the data to differentiate between heterozygosity and non-clonal infections, and by the careful analysis of LD decay with distance between markers, taking into account sample size and population structure. Overall, this is a masterclass in analysis methodology and certainly fulfills eLife's criteria that "Articles must be methodologically and scientifically rigorous, ethically conducted, and objectively presented according to the appropriate community standards."By examining samples from across the range of these species, the analysis clearly reveals fundamental differences between L.d. and L.i. and between different population samples of these parasites. While L.d. shows strong geographical structure, this is not the case for L.i. where closely related parasites are distributed across different continents. Similarly, some populations are genetically diverse and show strong evidence for hybridization and recombination (e.g. Turkey), other populations (e.g. Indian L.d.) show minimal diversity, and signatures of purely clonal propagation.The analysis of copy number and ploidy levels is extremely thorough. I was particularly interested that 28% of CNVs were found in both species, suggesting that either these are old shared polymorphisms or that they have been selected independently in the two species. Resolving the breakpoints (perhaps by searching for paired end reads that cross chromosome breakage sites) could help differentiate between these alternatives and also provide some clues as to the sequence motifs involved in chromosome breakage and copy number change, and provide some novelty to this study (see notes below).

Resolving the breakpoints on a large scale with short read data as used in this study is very challenging as broken, i.e. improper read pairs, mapped to the reference are generally dominated by noise (data not shown). In particular for the deletion on chromosome 27 (deletion 150) we did not find a single read pair spanning the deletion.We therefore investigated the breakpoint regions of the common deletion on chromosome 27 (deletion 150) and a few large duplications at the 5’ end of chromosome 35 in more detail by looking at the read coverage on a per bp resolution in the respective regions (Figure 7—figure supplement 7). This suggested identical breakpoints for deletion 150 and duplication 220 but differences between isolates for duplication 215 (Figure 7—figure supplement 7). We further searched for novel and previously described repeated sequences around the breakpoints of the respective CNVs (as also suggested by reviewer 1; Figure 7—figure supplement 7, Supplementary file 13). For the chromosome 27 deletion we could not find matching direct or inverted repeat sequences – as would be important for previously proposed mechanisms for generating copy number variation (Ubeda et al., 2014). However, repeated sequences were present near some of the breakpoint borders (Figure 7—figure supplement 7A), which can generally also interfere with the identification of read pairs supporting certain breakpoints. For the chromosome 27 deletion, identical breakpoints across isolates and the lack of matching repeat sequences on both sides of the deletion might suggest that this is an ancient segregating deletion in both species. We have now included these Results (Figure 7—figure supplement 7, Supplementary file 13).

I also like the hypothesis that rapid change in ploidy prevented accumulation of heterozygous sites during clonal division, although tests of this hypothesis were inconclusive.Weak featuresWhile the article is extremely rich in detail, I felt that it only incrementally increases our knowledge of leishmania population genomics, over previous work published by these and other authors. Previous work has demonstrated evidence for hybridization and recombination in both the laboratory and some field populations, has shown that Indian L.d. are extremely closely related, documented that mosiac aneuploidy is a central feature of leishmania, and identified strong association between copy number variants and drug resistance. I came away thinking that this is an impressive analysis, but wondering what were the key novel findings from this work – perhaps these could be more clearly highlighted in the Discussion?

We now more clearly highlight our novel findings throughout and at the end of the Discussion.

I was unsure why only a small subset of the published sequences were included in this analysis. For example, 46 sequences are available from Brazil but were not included, while only a 6 of 41 available from Ethiopia were included. There may be good reason for omission of samples (data not made available, or failed read depth cutoffs etc), but the manuscript would be improved by some rigid criteria for sample inclusion.

We aimed to include parasite strains from published sources in our analysis that added the already known genetic as well as geographic diversity that was otherwise not present in our newly sequenced isolates. Therefore, for regions, where the genetic diversity had already been described and many samples are available, we only took subsets of sequenced isolates: we took 2 samples from each of the 9 previously described clades from the Indian subcontinent (Imamura et al., 2016) and 6 samples from the two diverse lineages identified in Ethiopia (note: only 16 of the 41 samples come from different patients, Zackay et al., 2018). While previous studies using Brazilian isolates already suggest relatively little SNP-based diversity in South America, we should nevertheless have included some additional isolates to bring that original observation in the broader context of globally distributed isolates. Therefore, we have now added an additional phylogenetic analysis including the 26 Brazilian isolates from Carnielli et al., 2018, that present the greater sampling distribution of both Brazilian studies (i.e. Teixeira et al., 2017; Carnielli et al., 2018) to confirm the relatively little genetic diversity within South America and the monophyly of the South American “MON-1” lineage. Our additions are in the Result, Discussion and Material and methods sections (Figure 1—figure supplement 2).